# Leveraging plastomes for comparative analysis and phylogenomic inference within Scutellarioideae (Lamiaceae)

**Fei Zhao[1,2⊚], Bo Li[3⊚], Bryan T. Drew[4], Ya-Ping Chen[1], Qiang Wang[5], Wen-Bin Yu[6], En-De Liu[1], Yasaman Salmaki[7], Hua Peng[1]\*, Chun-Lei Xiang[1]\***

**1** CAS Key Laboratory for Plant Diversity and Biogeography of East Asia, Kunming Institute of Botany, Chinese Academy of Sciences, Kunming, China, **2** University of Chinese Academy of Sciences, Beijing, China, **3** Research Centre of Ecological Sciences, College of Agronomy, Jiangxi Agricultural University, Nanchang, China, **4** Department of Biology, University of Nebraska at Kearney, Kearney, Nebraska, United States of America, **5** State Key Laboratory of Systematic & Evolutionary Botany, Institute of Botany, Chinese Academy of Sciences, Xiangshan, Beijing, China, **6** Center for Integrative Conservation, Xishuangbanna Tropical Botanical Garden, Chinese Academy of Sciences, Mengla, China, **7** Center of Excellence in Phylogeny of Living Organisms and Department of Plant Sciences, School of Biology, College of Science, University of Tehran, Tehran, Iran

⊚ These authors contributed equally to this work.
\* hpeng@mail.kib.ac.cn (HP); xiangchunlei@mail.kib.ac.cn (CLX)

**Data Availability Statement:** All sequences used in this study are available from the National Center for Biotechnology Information (NCBI) MN128378–MN128389.

## Abstract

*Scutellaria*, or skullcaps, are medicinally important herbs in China, India, Japan, and elsewhere. Though *Scutellaria* is the second largest and one of the more taxonomically challenging genera within Lamiaceae, few molecular systematic studies have been undertaken within the genus; in part due to a paucity of available informative markers. The lack of informative molecular markers for *Scutellaria* hinders our ability to accurately and robustly reconstruct phylogenetic relationships, which hampers our understanding of the diversity, phylogeny, and evolutionary history of this cosmopolitan genus. Comparative analyses of 15 plastomes, representing 14 species of subfamily Scutellarioideae, indicate that plastomes within Scutellarioideae contain about 151,000 nucleotides, and possess a typical quadripartite structure. In total, 590 simple sequence repeats, 489 longer repeats, and 16 hyper-variable regions were identified from the 15 plastomes. Phylogenetic relationships among the 14 species representing four of the five genera of Scutellarioideae were resolved with high support values, but the current infrageneric classification of *Scutellaria* was not supported in all analyses. Complete plastome sequences provide better resolution at an interspecific level than using few to several plastid markers in phylogenetic reconstruction. The data presented here will serve as a foundation to facilitate DNA barcoding, species identification, and systematic research within *Scutellaria*, which is an important medicinal plant resource worldwide.

## Introduction

Lamiaceae is the sixth largest angiosperm family and contains over 7000 species that are divided into 12 subfamilies [1, 2]. Scutellarioideae, while relatively small, is one of the most morphologically distinct subfamilies within Lamiaceae. As circumscribed in earlier classifications [3, 4], the

**Funding:** This study was funded by the "Ten Thousand Talents Program of Yunnan (Top-notch Young Talents)" (No. YNWR-QNBJ-2018-279), CAS "Light of West China" Program and the "Excellent Youth Fund Project" (No. 2019FI009) of Yunnan Provincial Science and Technology Department to CLX, and the National Natural Science Foundation of China (No. 31870181) to QW.

**Competing interests:** The authors have declared that no competing interests exist.

subfamily contained only three genera, *Scutellaria* L., *Perilomia* Kunth, and *Salazaria* Torr., with the latter two genera synonymized with *Scutellaria* by Paton [5]. Subsequent studies based on morphological [6, 7] and molecular data [8, 9] expanded the subfamily to include *Renschia* Vatke, *Tinnea* Kotschy ex Hook. f., *Holmskioldia* Retz., and *Wenchengia* C. Y. Wu & S. Chow. Morphological synapomorphies for Scutellarioideae include pericarps with tuberculate or elongate processes [9], high densities of xylem fibers in the calyces [10], and racemose inflorescences (but most species of *Tinnea* and *Holmskioldia* have cymose inflorescences). The monophyly of the subfamily has also been supported by molecular phylogenetic studies [1, 8, 9, 11].

As currently defined, Scutellarioideae includes approximately 380 species in five genera [1]: *Holmskioldia*, *Renschia*, *Wenchengia*, *Scutellaria*, and *Tinnea*. The former three are monotypic genera. The genus *Holmskioldia*, comprising the single species *H. sanguinea* Retz., is native to the subtropical Himalayan region but is widely grown as an ornamental in warm climates and has become naturalized throughout the Old and New Worlds [12]. The monotypic *Renschia*, represented by *R. heterotypica* (S. Moore) Vatke, is narrowly endemic to the Ahl Mountains in northern Somalia [13], and its systematic position within Scutellarioideae remains unclear. The placement of *Wenchengia* in Scutellarioideae was resolved by Li et al. [9] based on the rediscovery of the extremely rare species, *W. alternifolia* C.Y. Wu & S. Chow. This genus was long thought to be endemic to Hainan Island in southern China [14, 15], but recently it was also reported from Vietnam [16]. With 19 species recognized to date, *Tinnea* is the second largest genus in Scutellarioideae, occurs mainly in fire-prone grassland, woodland, and scrub vegetation, and is endemic to Africa [17].

*Scutellaria*, containing approximately 360 species and commonly known as skullcaps, is the largest genus in Scutellarioideae [18]. The genus is distributed nearly worldwide and occurs in various habitats, but is mostly found in tropical montane and temperate regions [5, 19]. Most species are herbaceous perennials or small shrubs. The calyx of *Scutellaria* consists of two undivided lips and bears an appendage on the upper lip, which is described as a scutellum and is the most distinctive character of the genus; this feature is the basis for the common name skullcap. Many *Scutellaria* species possess medicinal uses, and some species are of economic importance. For example, *S. baicalensis* Georgi (baical skullcap or Chinese skullcap; 'Huang-qin' in Chinese) is a traditional Chinese medicinal herb that was first recorded in *Shen Nong Ben Cao Jing* in ca. 100 BC [20], and is widely used to treat hepatitis, jaundice, tumor, leukemia, hyperlipaemia, arteriosclerosis, diarrhea, and inflammatory diseases [21].

Due to tremendous diversity in habit, as well as calyx, corolla, inflorescence, and nutlet morphology, infrageneric boundaries within *Scutellaria* are poorly defined [3–5, 22–24]. Based on morphological data, Paton [5] subsumed *Harlanlewisia* Epling, *Perilomia*, and *Salazaria* into a broad *Scutellaria* as part of a global taxonomic revision, and divided *Scutellaria* into two subgenera: subg. *Scutellaria* and subg. *Apeltanthus* (Nevski ex Juz.) Juz. The former is further subdivided into five sections: *Scutellaria*, *Salviifoliae* (Boiss.) Edmondson, *Salazaria* (Torrey) Paton., *Perilomia* (Kunth) Epling, and *Anaspi* (Rech.f.) Paton. And the latter is divided into two sections: *Apeltanthus* and *Lupulinaria* A. Hamilt. As opposed to other large genera of Lamiaceae, such as *Plectranthus* L'Hér. [25–27], *Salvia* L. [28–34], and *Isodon* (Schrad. ex Benth.) Spach [35–38], molecular phylogenetic studies within *Scutellaria* are relatively scarce. Most previous work concentrated on genetic diversity and biogeography of taxonomic complexes (e.g. *S. angustifolia* Pursh [39, 40]), population genetics [21, 41], or species identification [42]. To date, only three phylogenetic studies have focused on *Scutellaria* [18, 41, 43]. Using both nuclear and chloroplast (cp) DNA markers, Chiang et al. [41] studied the relationships of Taiwanese *Scutellaria* and Safikhani et al. [18] focused on Iranian taxa. Similarly, when describing *S. wuana* C. L. Xiang & F. Zhao, only 41 taxa were involved in the phylogenetic analyses [43]. In total, only five DNA markers were used in these studies (nrITS, *matK*, *ndhF-*

*rpl32*, *rpl32-trnL*, and *trnL-trnF*) and none generated phylogenetic trees with high resolution, ostensibly due to a lack of variability within these DNA markers among the sampled species.

The chloroplast is an essential organelle in angiosperms because it provides energy for plant cells [44]. This uniparentally inherited plastid is characterized by a circular double-stranded DNA molecule between 120,000–160,000 base pairs in length, multiple copies per cell, and a quadripartite structure that includes two identical regions in opposite orientations called the inverted repeat (IR), flanked by large single copy (LSC) and small single copy (SSC) regions [45]. With increasingly rapid and less expensive next generation sequencing (NGS) technologies continually developing, ever-increasing numbers of non-model species plastid genome are being sequenced and successfully used for resolving phylogenetic and taxonomic problems in flowering plants at various ranks [46–48]. However, using cp genomes to resolve phylogenetic questions within the mint family has been rare [49], and plastomes of only two species, *Scutellaria baicalensis* and *S. indica* L. var. *coccinea* S. Kim & S. Lee, have been published from Scutellarioideae [50, 51]. Sequences of *S. insignis* Nakai and *S. lateriflora* L. were uploaded to GenBank without any related publication or analyses. Consequently, little is known regarding plastome structure variation within *Scutellaria*.

In this study, we sequenced 12 plastomes from 11 species representing four of the five genera of Scutellarioideae. In addition, three previously released plastomes of *Scutellaria* (*S. baicalensis*, *S. insignis* and *S. lateriflora*) were downloaded from GenBank and included for comparative analyses. The species *S. indica* var. *coccinea* was exclude in this study because the sequence was unavailable. With these data, we aim to: 1) characterize and compare the structure and gene organization of plastid genomes within Scutellarioideae; 2) identify candidate molecular markers for future phylogenetic and/or population genetic studies within *Scutellaria*; and 3) reconstruct the phylogeny of Scutellarioideae using complete chloroplast genome sequences. The data presented in this study will provide abundant information for further studies about phylogeny, taxonomy, species identification, and population genetics of *Scutellaria*, and will also be helpful for exploration, utilization, and conservation of plant genetic resources of this important medicinal plant resources.

## Materials and methods

### Taxon sampling, DNA extraction, and sequencing

Plastomes of 12 samples, including eight species of *Scutellaria*, one species each of *Holmskioldia* and *Tinnea*, and two individuals of *Wenchengia alternifolia*, were newly generated for this study. Voucher information is listed in Table 1 and all voucher specimens were deposited at the Herbarium of Kunming Institute of Botany (KUN), Chinese Academy of Sciences. In addition, three complete plastomes of *Scutellaria* from GenBank, *S. baicalensis* (MF521633), *S. insignis* (KT750009), and *S. lateriflora* (KY085900), were included for comparative analyses (Table 1).

Total genomic DNA was extracted from 150 mg fresh or silica-gel dried leaves using the CTAB method [52]. The DNA samples were sheared into fragments of about 300 bp to construct libraries according to manufacturer's instructions (Illumina, San Diego, CA, USA). Paired-end (PE) sequencing of 150 bp was conducted on an Illumina Hiseq-2500 platform (Illumina Inc.) at BGI-Wuhan.

Quality control of raw sequence reads was carried out using FastQC toolkit (http://www.bioinformatics.babraham.ac.uk/projects/fastqc; [53]) with the parameter set as Q ≥ 25 to acquire high-quality clean reads for downstream analyses. *De novo* assembling of the plastomes was implemented in the GetOrganelle pipeline [54]. The filtered *de* Bruijn graphs file "gfa" was visualized in Bandage v. 0.8.1 [55] and the complete chloroplast sequence paths were

**Table 1. Voucher information of the newly sequenced samples in this study.**

| Species | Location | Vouchers | Coordinate |
|---|---|---|---|
| *Wenchengia alternifolia* C.Y. Wu & S. Chow HN | China, Hainan, Ding'an | Xiang et al. 1318 | E 110°17′50.19″, N 19°13′53.27″ |
| *W. alternifolia* C.Y. Wu & S. Chow VN | Vietnam, DaNang, Ba na Hill | Li et al. Lbo824 | E 107°59′17.91″, N 15°59′59.80″ |
| *Holmskioldia sanguinea* Retz. | China, Yunnan, XTBG* | Zhao et al. ZF014 | E 101°15′21.10″, N 21°55′38.06″ |
| *Tinnea aethiopica* Kotschy ex Hook. f. | Kenya, Kabarnet | Li et al. 4292 | E 35°44′36.30″, N 0°29′24.38″ |
| *Scutellaria amoena* var. amoena C.H. Wright | China, Yunnan, Kunming | Zhao et al. ZF034 | E 102°43′07.74″, N 25°07′19.91″ |
| *S. calcarata* C.Y. Wu & H.W. Li | China, Yunnan, Gongshan | Li et al. NJ023 | E 98°39′39.55″, N 27°44′26.32″ |
| *S. mollifolia* C.Y. Wu & H.W. Li | China, Sichuan, Emei | Chen et al. EM201 | E 103°20′01.25″, N 21°55′38.06″ |
| *S. orthocalyx* Hand.-Mazz. | China, Yunnan, KBG* | Zhao et al. ZF035 | E 102°44′38.26″, N 25°08′27.10″ |
| *S. quadrilobulata* Y.Z. Sun | China, Yunnan, Xinping | Li et al. XP965 | E 101°56′55.70″, N 23°56′51.32″ |
| *S. kingiana* Prain | China, Xizang, Cuona | Yang et al. ZJW-3890 | E 99°56′09.26″, N 28°05′18.95″ |
| *S. altaica* Fisch. ex Sweet | China, Xinjiang, Xinyuan | Zhang et al. 17CS16318 | E 84°02′23.18″, N 43°18′24.83″ |
| *S. przewalskii* Juz. | China, Xinjiang, Aletai | Chen et al. YC_ZX027 | E 88°02′42.31″, N 47°20′41.62″ |

*: XTBG: Xishaungbanna Tropical Botanical Garden; KBG: Kunming Botanical Garden.

manually selected, with the minimum depth of contigs above 100 × and the minimum length > 300 bp. Then all PE reads were mapped to the assembled plastomes using the Bowite2 [56] plugin in Geneious v.11.0.4 [57] to verify quality and correct assembly errors.

Plastome annotation was first performed using the online programs Dual Organellar Genome Annotator (DOGMA) [58] and Ge-seq [59]. We then inspected and curated all annotation manually with comparisons to the published plastome of *S. baicalensis* (MF521633) in Geneious v.11.0.4 [57]. The tRNAs were verified using the online tRNAscan-SE service with default parameters [60]. The resulting circular plastome maps were drawn using the OrganellarGenomeDRAW tool [61].

## Characterization of simple sequence repeats and repeat structure

The simple sequence repeats (SSRs) in plastomes were identified using MISA perl script (http://pgrc.ipk-gatersleben.de/misa). Thresholds for the minimum repeated size were set as follows: ≥ 10 for mono-nucleotide, ≥ 5 for di-nucleotide, ≥ 4 for tri-nucleotide, and ≥ 3 for tetra-nucleotide, penta-nucleotide, and hexa-nucleotide repeats. The location and size of the repeating sequences (forward, reverse, palindromic and complement) were visualized in REPuter [62] with the parameter set as with a hamming distance of 3 and a minimum repeat size of 30 bp following the procedure outlined in Jiang et al. [50].

## Comparative plastome and sequence divergence analysis

Comparative analyses of 15 plastomes of Scutellarioideae were carried out using the Mauve v.2.3.1 [63] plugin in Geneious v.11.0.4 [57]. We applied mVISTA [64] to visualize the results and evaluate the similarity among different plastomes, using default parameters to align plastomes under the LAGAN model and the annotations of *S. baicalensis* (MF521633) as a reference. In order to investigate the IR contraction or expansion, we also compared the boundaries between IR and SC regions in Geneious v.11.0.4 [57]. Two data sets (alignments of all 15 samples from Scutellarioideae and 11 species of *Scutellaria*) were used for the sliding window analysis to evaluate the intergeneric and intrageneric nucleotide sequence variabilities (Pi). Sequences were aligned using MAFFT v.7.221 [65] and misaligned regions were manually adjusted in Geneious v.11.0.4. [57]. DnaSP v.6 [66] was then used to calculate the Pi. The step size was set to 200 bp, with a 600 bp window length.

**Table 2. Features of the complete plastomes of 15 species of Scutellarioideae.** (NA means not available).

| Taxa | Accession number | Complete | | LSC | | SSC | | IR | | Assembly Reads | Mean coverage (×) | Genes number | Protein coding genes | tRNA genes | rRNA genes |
|---|---|---|---|---|---|---|---|---|---|---|---|---|---|---|---|
| | | Length (bp) | GC content (%) | Length (bp) | GC content (%) | Length (bp) | GC content (%) | Length (bp) | GC content (%) | | | | | | |
| *Holmskioldia sanguinea* | MN128389 | 153,272 | 38.20 | 84,688 | 36.30 | 17,330 | 32.50 | 25,627 | 43.40 | 27,007,418 | 1655 | 114 | 80 | 30 | 4 |
| *Wenchengia alternifolia* HN | MN128379 | 152,843 | 38.30 | 84,807 | 36.30 | 16,768 | 32.80 | 25,634 | 43.40 | 21,196,548 | 1943 | 114 | 80 | 30 | 4 |
| *W. alternifolia* VN | MN128378 | 152,171 | 38.30 | 84,329 | 36.30 | 16,750 | 32.70 | 25,546 | 43.40 | 24,616,880 | 3068 | 114 | 80 | 30 | 4 |
| *Tinnea aethiopica* | MN128380 | 152,450 | 38.40 | 84,414 | 36.40 | 17,482 | 32.60 | 25,277 | 43.60 | 25,813,343 | 1612 | 114 | 80 | 30 | 4 |
| *Scutellaria altaica* | MN128387 | 151,779 | 38.30 | 83,984 | 36.30 | 17,327 | 32.60 | 25,234 | 43.60 | 24,309,408 | 618 | 114 | 80 | 30 | 4 |
| *S. amoena* var. *amoena* | MN128386 | 151,833 | 38.30 | 84,001 | 36.30 | 17,340 | 32.70 | 25,246 | 43.60 | 21,404,966 | 2893 | 114 | 80 | 30 | 4 |
| *S. calcarata* | MN128385 | 152,033 | 38.40 | 84,023 | 36.40 | 17,532 | 32.60 | 25,239 | 43.60 | 24,676,179 | 3257 | 114 | 80 | 30 | 4 |
| *S. kingiana* | MN128388 | 152,395 | 38.30 | 84,608 | 36.30 | 17,305 | 32.40 | 25,241 | 43.60 | 22,854,411 | 4510 | 114 | 80 | 30 | 4 |
| *S. mollifolia* | MN128384 | 152,417 | 38.30 | 84,432 | 36.40 | 17,569 | 32.60 | 25,208 | 43.60 | 25,602,286 | 1755 | 114 | 80 | 30 | 4 |
| *S. orthocalyx* | MN128383 | 152,071 | 38.40 | 84,072 | 36.40 | 17,519 | 32.60 | 25,240 | 43.60 | 16,687,912 | 1577 | 114 | 80 | 30 | 4 |
| *S. przewalskii* | MN128382 | 151,675 | 38.30 | 83,891 | 36.40 | 17,320 | 32.60 | 25,232 | 43.60 | 24,156,080 | 1496 | 114 | 80 | 30 | 4 |
| *S. quadrilobulata* | MN128381 | 152,066 | 38.30 | 84,052 | 36.40 | 17,544 | 32.50 | 25,235 | 43.60 | 25,241,479 | 2740 | 114 | 80 | 30 | 4 |
| *S. baicalensis* | MF521633 | 151,817 | 38.30 | 83,960 | 36.30 | 17,331 | 32.70 | 25,263 | 43.60 | NA | NA | 114 | 80 | 30 | 4 |
| *S. insignis* | KT750009 | 151,908 | 38.40 | 83,913 | 36.50 | 17,517 | 32.60 | 25,239 | 43.60 | NA | NA | 114 | 80 | 30 | 4 |
| *S. lateriflora* | KY085900 | 152,283 | 38.30 | 84,340 | 36.30 | 17,465 | 32.50 | 25,239 | 43.60 | NA | NA | 114 | 80 | 30 | 4 |

### Phylogenetic analysis based on complete plastome sequences

In addition to the previously published plastomes of *Scutellaria*, plastomes of 31 species from within other subfamilies of Lamiaceae (12 Nepetoideae, 15 Lamioideae, two Ajugoideae, and one each from Premnoideae and Tectonoideae) were also included in the analyses to evaluate the utility of complete plastome sequences for resolving broad relationships within Scutellarioideae. Based on previous studies [1], *Callicarpa americana* (assembly from the WGS data under the SRR6940059) from Callicarpoideae was selected as the outgroup. GenBank accession numbers are provided in S1 Table.

Alignments were initially performed using MAFFT v.7.221 [65] with default settings, and subsequently manually adjusted in Geneious v.11.0.4 [57]. Ambiguously aligned regions (e.g. characters of uncertain homology among taxa and single-taxon insertions) were excluded before phylogenetic analyses. Since the plastid genome is uniparentally inherited and does not undergo recombination [67], we combined all sequences and constructed three matrices: (i) combined coding regions (dataset CR); (ii) combined non-coding regions (dataset NCR); (iii) combined whole plastome sequences (dataset CPG). In order to reduce the overrepresentation of duplicated sequences, only the IRa region was included in all data sets. In addition, in order to evaluate the efficacy of the complete plastome sequences for phylogeny reconstruction within Scutellarioideae, we also created two additional datasets for phylogenetic analyses and comparison. One was a combined dataset of hyper-variable regions (16VAR) detected in this study, the other dataset consisted of six commonly used DNA regions (6CP) from previous studies [9, 41, 68].

Maximum likelihood (ML) and Bayesian inference (BI) analyses were performed on the Cyberinfrastructure for Phylogenetic Research Science (CIPRES) Gateway (http://www.phylo.org/; [69]. ML analyses were conducted using RAxML HPC2 v.8.2.9.0 [70] with the general time reversible (GTR) + G model and 1000 bootstrap replicates. BI analyses were carried out using MrBayes v.3.2.6 [71]. The best substitution model for each data set was determined using jModelTest2 [72] on the CIPRES Gateway, under the Bayesian information criterion (BIC) [73]. Four Markov Chain Monte Carlo (MCMC) chains (one cold and three heated) were run for 20 million generations. Convergence of the MCMC runs and estimated sample size (ESS) were analyzed by Tracer v.1.7.0 [74]. The first 25% of trees discarded as burn-in, and the remaining trees were summarized to construct the 50% majority-rule consensus tree.

## Results

### Genome assembly, features, and gene content across scutellarioideae

Illumina paired-end sequencing generated 16,687,912–27,007,418 clean reads for the 12 newly sequenced samples, with the mean coverage ranging from 618× in *Scutellaria altaica* Fisch. ex Sweet to 4510× in *S. kingiana* Prain. The genome size ranged from 151,675 bp in *S. przewalskii* Juz. to 153,272 bp in *Holmskioldia sanguinea* (Table 2). All 15 plastomes of Scutellarioideae displayed the typical quadripartite structure consisting of a pair of IR regions (25,208–25,634 bp) separated by the LSC (83,891–84,807bp) and SSC (16,750–17,569 bp) regions (Table 2). The GC content was similar among different species of Scutellarioideae and the average GC content was 38.3% (Table 2). In general, the GC content in the IR regions (43.4–43.6%) was higher than in the LSC (36.3–36.5%) and SSC (32.4–32.8%) regions, and the GC content within non-coding regions (35.0%) was lower than within coding regions (40.5%).

Intraspecific plastome polymorphisms can be evaluated among multiple individuals from the same species. The sequence identity between the two samples of *Wenchengia alternifolia* was 98.6%, with only two large indels (> 100 bp), within the intergenic *psbE-petL* (344 bp) and

*psbM-trnD (GUC)* (226 bp) regions, detected. The plastome maps of *Holmskioldia sanguinea*, *W. alternifolia* HN, *Tinnea aethiopica*, and *Scutellaria przewalskii* are presented as representatives of Scutellarioideae (Fig 1), while maps of the remaining species are provided in supplementary materials (S1 Fig). All newly sequenced and annotated plastomes were submitted to the National Center for Biotechnology Information (NCBI) database under accession numbers MN128378–MN128389 (Table 2).

When duplicated genes in IR regions were counted only once, each of the plastomes included 114 unique genes (80 protein-coding genes, 30 tRNAs and four rRNAs; Table 2) that

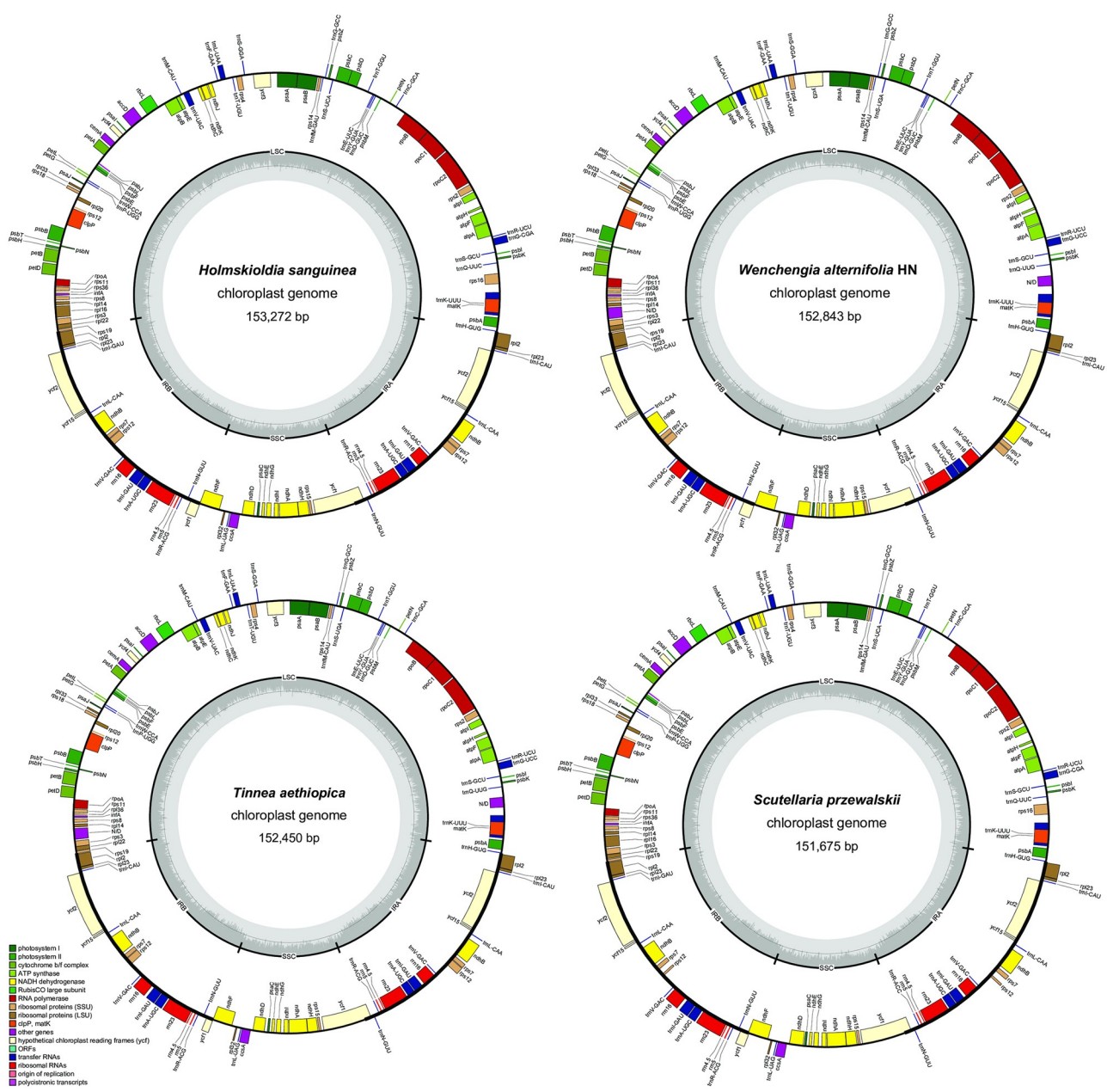

**Fig 1. Complete plastome maps of *Holmskioldia sanguinea*, *Wenchengia alternifolia*, *Tinnea aethiopica*, and *Scutellaria przewalskii*.**

were arranged in the same order. A total of 18 genes exist in duplication within the IR region, including seven protein-coding genes, seven tRNAs and four rRNAs (Table 3). Ten of the protein-coding genes and six of the tRNA genes contained one intron, and two genes (*ycf3* and *clpP*) contained two introns. Among those newly sequenced samples, protein-coding regions accounted for 52.1–53.5% of the length of the whole genome, while tRNA and rRNA regions accounted for 1.78–1.92% and 5.9–5.96%, respectively (S2 Table). The remaining regions were non-coding sequences, including intergenic spacers, introns, and pseudogenes. All of the gene functions and groups were shown in Table 3.

**Table 3. The gene functions of the plastomes of 15 species of Scutellarioideae.**

| Category for genes | Group of genes | Name of genes |
|---|---|---|
| Photosynthesis | Subunits of NADH-dehydrogenase | *ndhA*\*, *ndhB*\*(2x), *ndhC*, *ndhD*, *ndhE*, *ndhF*, *ndhG*, *ndhH*, *ndhI*, *ndhJ*, *ndhK* |
| | Photosystem I | *psaA*, *psaB*, *psaC*, *psaI*, *psaJ*, *ycf3*\*\* |
| | Photosystem II | *psbA*, *psbB*, *psbC*, *psbD*, *psbE*, *psbF*, *psbH*, *psbI*, *psbJ*, *psbK*, *psbL*, *psbM*, *psbN*, *psbT*, *psbZ* |
| | Cytochrome b/f complex | *petA*, *petB*\*, *petD*\*, *petG*, *petL*, *petN* |
| | ATP synthase | *atpA*, *atpB*, *atpE*, *atpF*\*, *atpH*, *atpI* |
| | Large chain of rubisco | *rbcL* |
| Self-replication | Ribosomal RNA genes | *rrn16* (2x), *rrn23* (2x), *rrn4.5* (2x), *rrn5* (2x) |
| | Transfer RNA genes 30 tRNA genes | (6 contain one intron, 7 are duplicated in the IR region) |
| | | *trnA-UGC*\*(2x), *trnfM-CAU*, *trnI-GAU*\*(2x), *trnM-CAU*, *trnR-ACG*(2x), *trnS-UGA*, *trnC-GCA*, *trnG-GCC*\*, *trnK-UUU*\*, *trnN-GUU(2x)*, *trnW-CCA*, *trnT-GGU*, *trnD-GUC*, *trnG-UCC*, *trnL-CAA(2x)*, *trnY-GUA*, *trnR-UCU*, *trnT-UGU*, *trnE-UUC*, *trnH-GUG*, *trnL-UAA*\*, *trnP-UGG*, *trnS-GCU*, *trnV-GAC(2x)*, *trnF-GAA*, *trnI-CAU(2x)*, *trnL-UAG*, *trnQ-UUG*, *trnS-GGA*, *trnV-UAC*\* |
| | Small subunit of ribosome | *rps2*, *rps3*, *rps4*, *rps7* (2x), *rps8*, *rps11*, *rps12*, *rps14*, *rps15*, *rps16*\*, *rps18*, *rps19* |
| | Large subunit of ribosome | *rpl2*\* (2x), *rpl14*, *rpl16*\*, *rpl20*, *rpl22*, *rpl23* (2x), *rpl32*, *rpl33*, *rpl36* |
| | RNA polymerase subunits | *rpoA*, *rpoB*, *rpoC1*\*, *rpoC2* |
| Other genes | Translation initiation factor | *infA* |
| | Maturase | *matK* |
| | Protease | *clpP*\*\* |
| | Envelope membrane protein | *cemA* |
| | Subunit of acetyl-CoA–carboxylase | *accD* |
| | cytochrome c biogenesis protein | *ccsA* |
| | Component of TIC complex | *ycf1* |
| Genes of unknown function | | *ycf2*, *ycf4*, *ycf15* (2x) |

\*gene with a single intron,

\*\*gene with two introns, (2x) duplicated gene.

## SSRs and repeat structure

In total, 590 SSRs were identified in the 15 plastomes of Scutellarioideae, of which 483 SSRs (81.86%) were in the LSC region, 65 SSRs (11.02%) were in the SSC region, and 42 SSRs (7.12%) were in the IR region (Fig 2, S3 Table). The number of SSRs (or microsatellite loci) ranged from 31 (*Scutellaria altaica*) to 48 (*Wenchengia alternifolia* HN) among species of Scutellarioideae (Fig 2). The mononucleotide represents the highest variability with the repeat number ranging from 15 (*S. altaica*) to 35 (*W. alternifolia* HN), while the number of dinucleotide, trinucleotide, and tetranucleotide repeats showed no significant difference among the 15 samples. The number and frequency of each repeat type within the 15 plastomes of Scutellarioideae is shown in Fig 2 and S3 Table.

When the cyclic queues and reverse complements were regarded as the same SSRs, the 590 SSRs can be classified into 17 different repeat types. The mononucleotide repeat unit (A/T); dinucleotide repeat unit (AT/AT), trinucleotide repeats unit (AAG/CTT) and tetranucleotide repeat unit (AAAG/CTTT, AAAT/ATTT) were shared in all the 15 samples (Fig 3). The mononucleotide repeat unit (G/C) was absent in *Scutellaria calcarata*. Within the trinucleotide repeat, the repeat unit (AAC/GTT) was unique to *Wenchengia*, and the repeat unit (AAT/ATT)

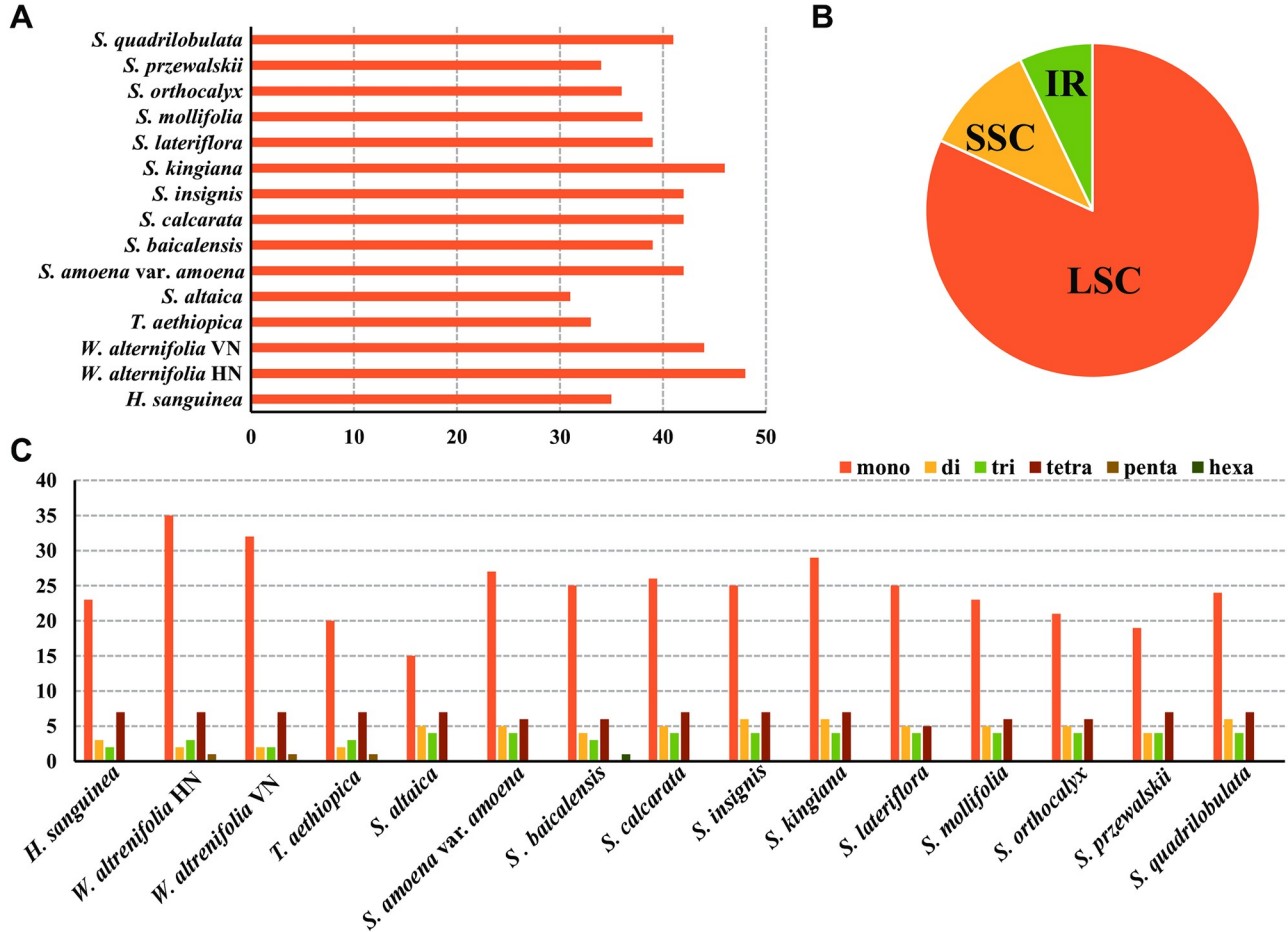

**Fig 2. Comparisons of the simple sequence repeats (SSR) among the 15 plastomes of Scutellarioideae. (A)** Number of SSRs detected in each plastome; **(B)** Frequencies of identified SSRs in LSC, IR, and SSC regions; **(C)** Number of SSR types detected in each plastome.

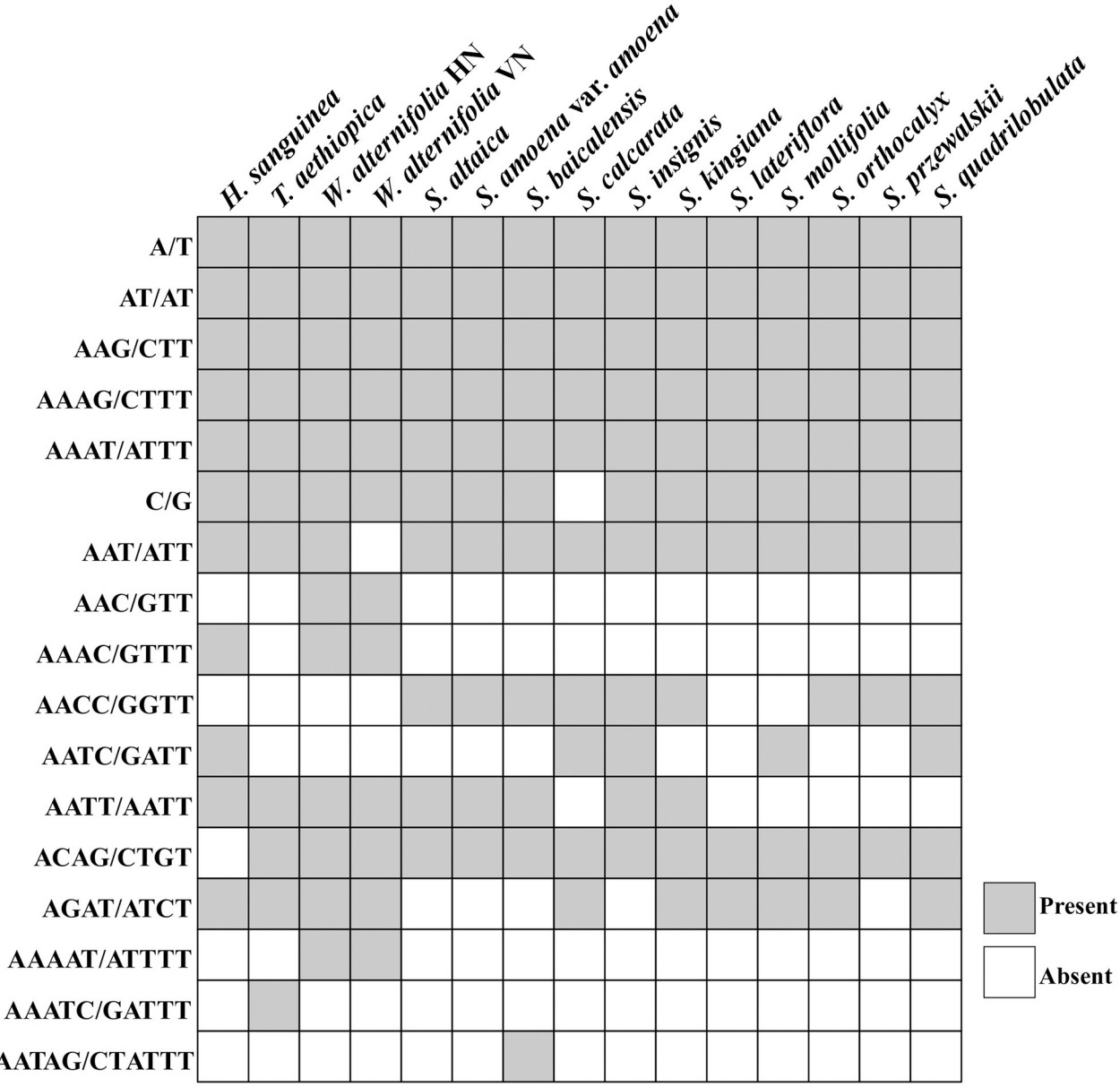

**Fig 3. Distribution of the 17 types of SSR repeat units among 15 plastomes of Scutellarioideae and their relationships.** The horizontal axis indicates the species name and the Y-scale indicates the type of repeat unit.

was shared by the other samples except the *Wenchengia alternifolia* VN accession. The tetranucleotide repeats showed the most polymorphisms, the repeat unit (AAAC/GTTT) were shared with *Holmskioldia* and the two samples of *Wenchengia*; the repeat unit (AACC/GGTT) was detected in nine species from *Scutellaria*; the repeat unit (AATC/GATT) was found in *Holmskioldia* and four *Scutellaria* species (*S. calcarata*, *S. insignis*, *S. mollifolia* and *S. quadrilobulata*); the repeat unit (AATT/AATT) was not found in *S. calcarata*, *S. lateriflora*, *S. mollifolia*, *S. orthocalyx* and *S. quadrilobulata*. The repeat unit (ACAG/CTGT) was shared by other species excluding the *Holmskioldia*, and repeat unit (AGAT/ATCT) didn't present in *S. altaica*, *S. amoena* var. *amoena*, *S. baicalensis*, *S. insignis* and *S. przewalskii*. The pentanucleotide repeats

were detected in both individuals of *W. alternifolia* and in *Tinnea aethiopica*, while the hexanu-cleotide repeats were only found in *S. baicalensis*. The distribution of the 17 repeat types among the 15 plastomes and their relationships is shown in Fig 3.

In total, 489 long repeats including forward, reverse, and palindromic were detected in the 15 plastomes (Fig 4). The most abundant type were the palindromic repeats, which accounted for 54.26% of the total repeats, followed by forward repeats (44.91%). The reverse repeats were rare and accounted for only 0.83% of the total repeats (Fig 4). Most repeats were located in the non-coding regions (77.96%; Fig 4). The length of the repeats ranged from 30 bp to 136 bp, and most of the repeat sequences were 30 bp, 32 bp, 39 bp, 41 bp, and 60 bp long (Fig 4, S4 Table).

## Comparative analysis of plastomes of Scutellarioideae

The Mauve results showed that the organization of the plastomes in Scutellarioideae is highly conserved; neither translocations nor inversions were detected. However, differences in the size of the plastomes were detected. For example, the plastome of *Scutellaria przewalskii* was the shortest (151,675 bp), while that of *Holmskioldia sanguinea* (153,272 bp) was longer than the other species (S2 Fig). Results from the analyses by mVISTA showed that the two IR regions were less divergent than the LSC and SSC regions. Moreover, the non-coding regions and the intergenic spacers exhibited a higher divergence than the coding regions (Fig 5). In all species, the IRa/LSC junctions were located within the *rps19* gene, with a 41–74 bp protrusion of the *rps19* gene into the IRa region that resulted in a part of the *rps19* gene (ψ*rps19*) present in the IRb region. In *Wenchengia alternifolia* and *Tinnea aethiopica*, the *ndhF* gene was completely located in the SSC region while in *H. sanguinea* and all species of *Scutellaria* a small fragment of the *ndhF* gene extended into the IRa region with (29 bp in *H. sanguinea* and 25–45 bp among species of *Scutellaria*). The IRb/SSC boundary was within the *ycf1* gene, with between 771 and 1,184 bp in the IRb region. An equal length *ycf1* pseudo-gene (ψ*ycf1*) was detected in the IRa region. The IRb/LSC boundary was located between the pseudogene *rps19* (ψ*rps19)* and *trnH-GUG* across the 15 plastomes. The distance between *trnH-GUG* and the IRb/LSC boundary for all species varied from 0 to 3 bp (Fig 6).

## Sequence divergence and nucleotide diversity

The average nucleotide variability (Pi) of plastomes was estimated to be 0.004 in *Scutellaria* (Fig 7). The SSC region showed the highest average nucleotide diversity (Pi = 0.0148), followed by the LSC region (Pi = 0.0087) and the IR region (Pi = 0.0019). Among the 11 species of *Scutellaria*, ten hyper-variable regions were identified, including two genes *(ndhF, ycf1)* and eight intergenic spacers (*psbA-trnH*, *trnK-rps16* intron, *petN-psbM*, *rbcL-accD*, *petA-psbJ*, *petB-petD* intron, *rpl32-trnL*, and *rps15-ycf1*), with the variation exceeding 2.0%.

As for the 15 samples of Scutellarioideae, the average nucleotide variability (Pi) of the whole plastome was 0.014, while that of the LSC, SSC, and IR regions were 0.0178, 0.028, and 0.003, respectively. In the LSC region, we found 11 hyper-variable loci with Pi values > 0.03 (*psbA-trnH*, *trnK-rps16* intron, *atpH-atpI*, *rpoB-trnC*, *petN-psbM*, *ycf3-trnS*, *trnT-trnF*, *rbcL-accD*, *ycf4-cemA*, *petA-psbJ*, and *petB-petD* intron), while in the SSC region, only five hyper-variable loci with Pi values > 0.03 (*ndhF*, *rpl32-trnL*, *ccsA-ndhD*, *rps15-ycf1*, and *ycf1*) were detected (Fig 7).

## Characteristics of the datasets and phylogenetic relationships within Scutellarioideae

After the exclusion of ambiguously aligned sites, the total length of the complete aligned data-set (CPG) was 144,120 bp, of which 36,934 bp were variable (25.63%). The length of the CR

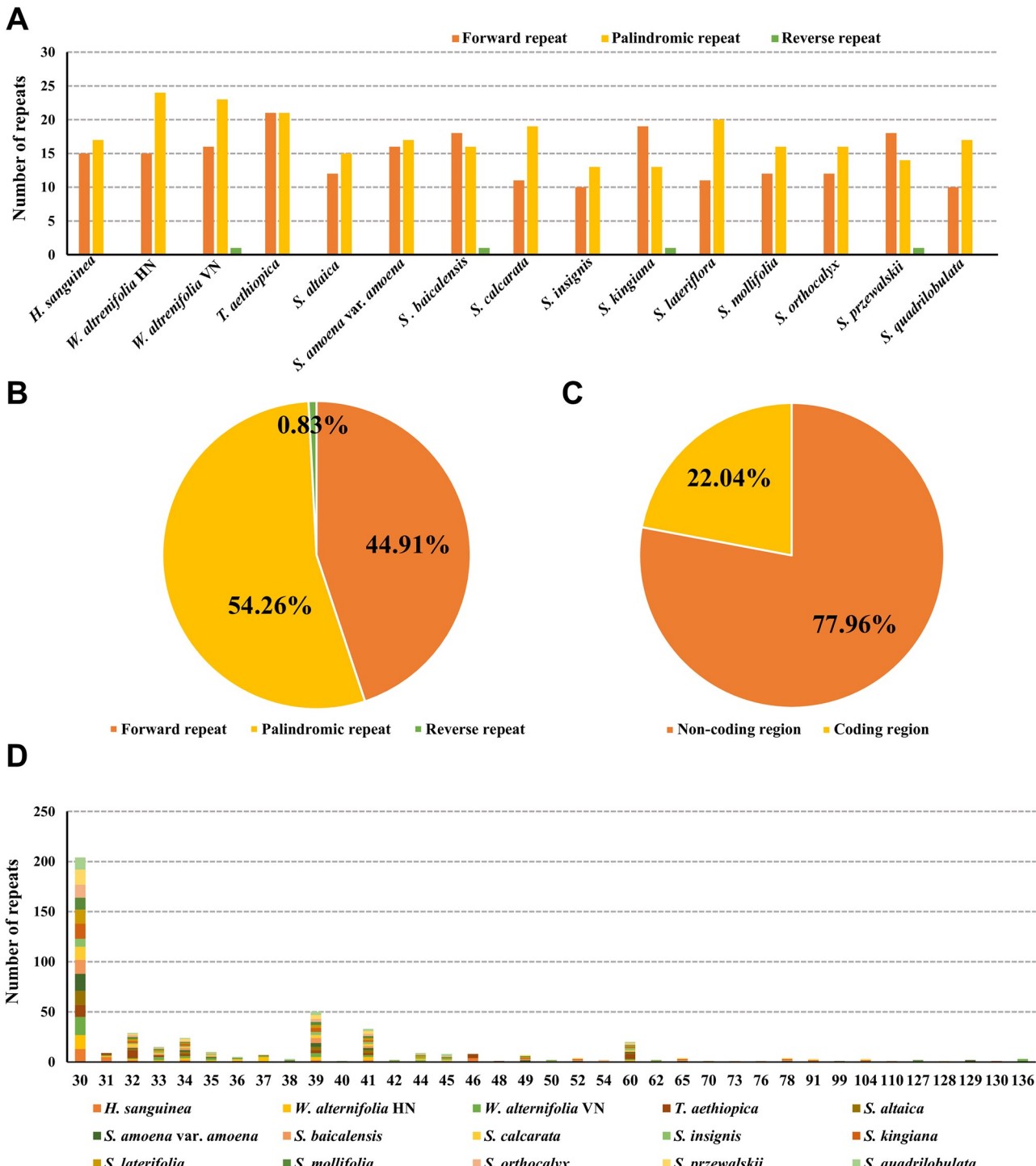

**Fig 4. Long repeat sequences in the complete plastomes of 15 taxa of Scutellarioideae. (A)** Number of repeat types detected in each plastome; **(B)** Frequency of each repeat type; **(C)** Percentages of repeat type loci in the non-coding and coding regions; **(D)** Frequencies of repeats longer than 30 bp.

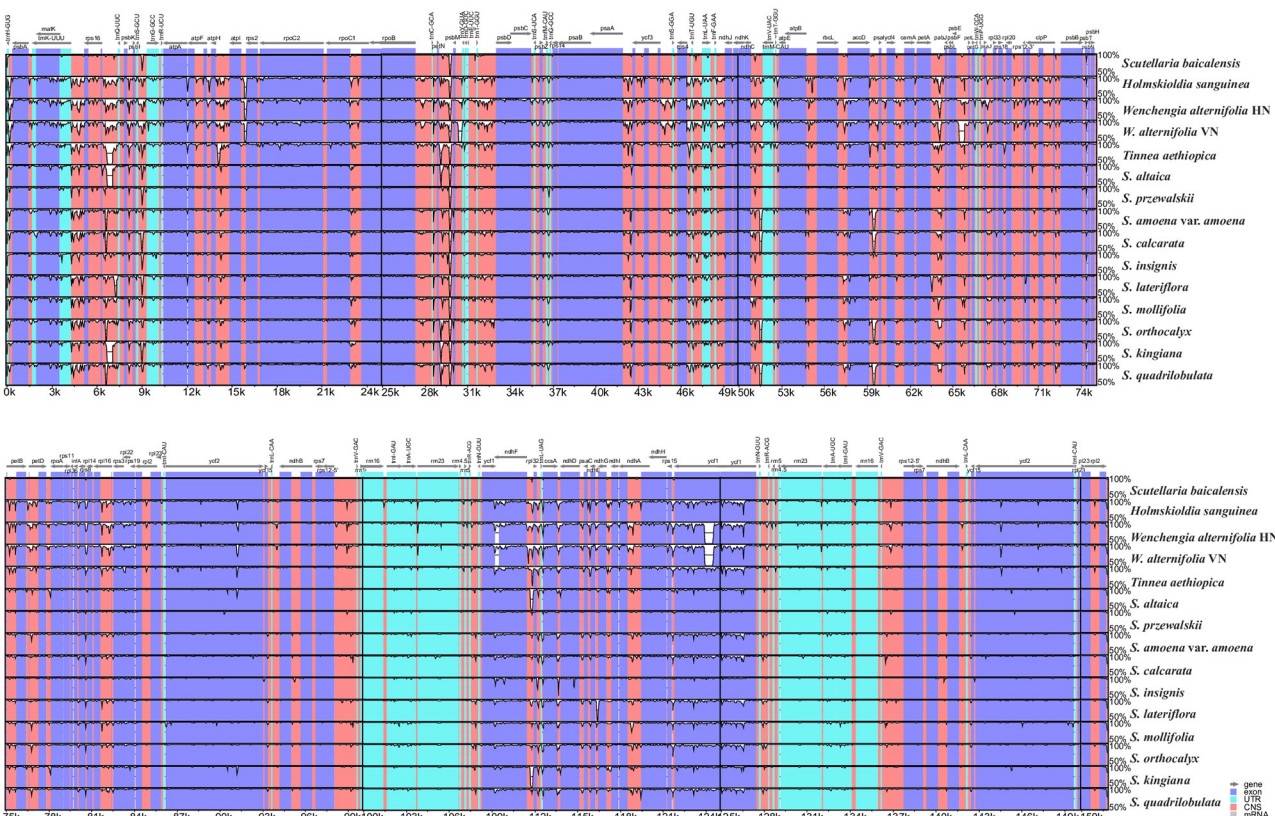

**Fig 5. Sequence alignment of the whole plastomes of 15 taxa of Scutellarioideae using the LAGAN alignment algorithm in mVISTA, with *Scutellaria baicalensis* as the reference.** The horizontal axis indicates the coordinates within the plastomes. The Y-scale indicates the percentage of identity, ranging from 50% to 100%. Genome regions are color coded as protein coding, trnA gene, rrnA gene, intron, mRNA, and conserved non-coding sequences.

dataset was 70,046 bp, of which 14,288 bp (20.4%) were variable. The noncoding dataset (NCR) was 72,624 bp, of which 23,032 bp (31.71%) were variable. The hyper-variable dataset (16VAR) was 24,090 bp, of which 9,953 bp (39.8%) were variable. The six commonly used cpDNA regions (6CP) was 8,346 bp, of which 2,820 bp (33.6%) were variable. Data characteristics with models selected for each dataset used for Bayesian phylogenetic analyses are list in Table 4. Topologies obtained from both ML and BI analyses for all three datasets were identical, thus the ML topology resulting from the analysis of the CPG dataset (Fig 8) is presented here for subsequent discussion of phylogenetic relationships.

In all our analyses, the Scutellarioideae was supported as monophyletic (ML/BS 100%, BI/PP 1.00) [all values follow this order hereafter] (Fig 8, S3–S8 Figs). The two samples of the monotypic genus *Wenchengia* formed a well-supported clade (100%, 1.00) sister to remaining genera of Scutellarioideae. All species of *Scutellaria* were recovered in a strongly supported clade (100%, 1.00), in which two subclades were recognized. Subclade I (100%, 1.00) comprised five species from three sections: sect. *Lupulinaria* (*S. altaica* and *S. przewalskii*, sect. *Scutellaria* (*S. baicalensis* and *S. amoena* var. *amoena*), and sect. *Anaspis* (*S. kingiana*). Subclade II (100%, 1.00) consist of six species from sect. *Scutellaria*.

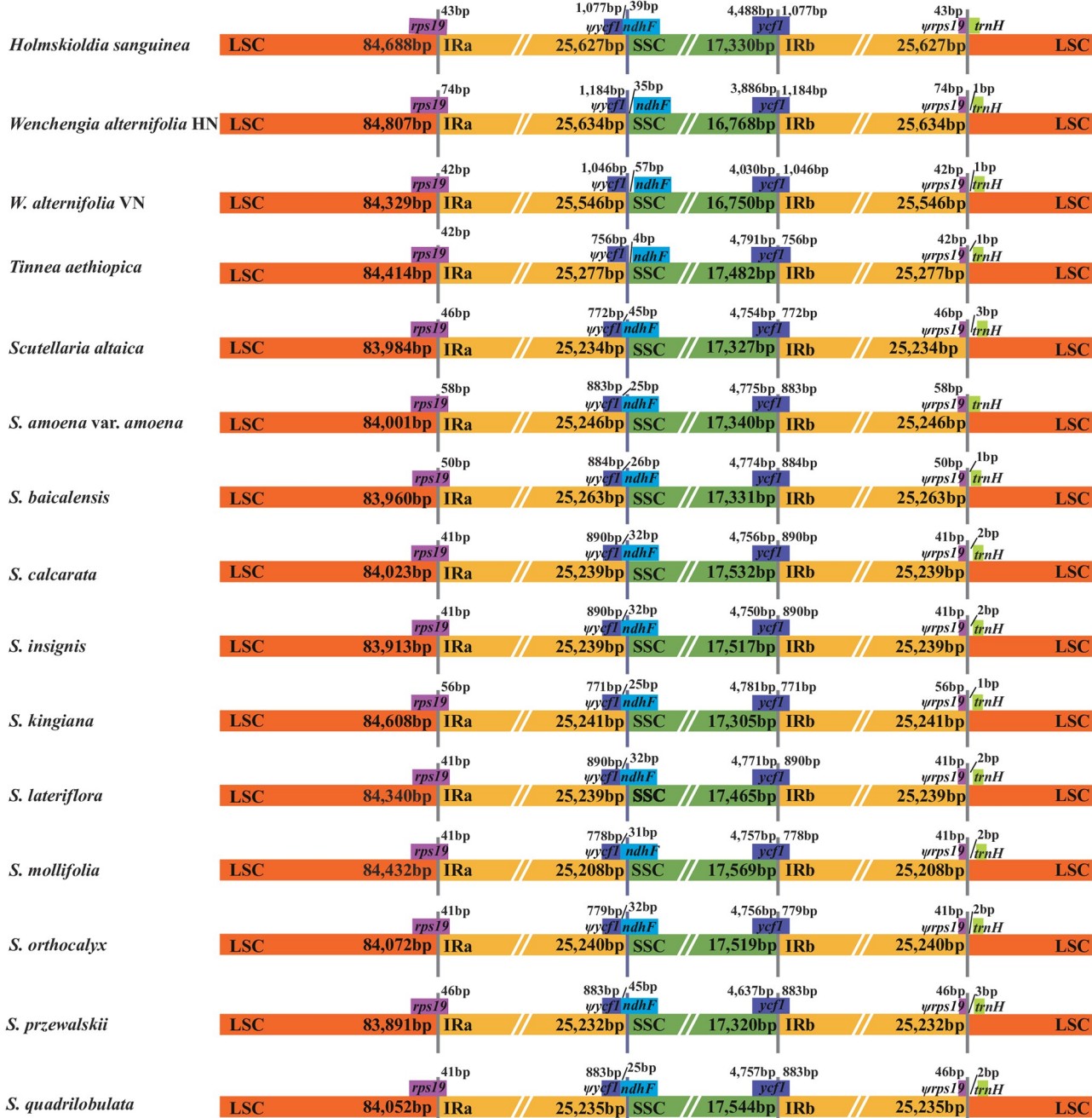

**Fig 6. Comparisons of the LSC, IR, and SSC borders of plastomes of *Scutellaria* and related genera.**

## Discussion

### General characteristics of the plastomes of Scutellarioideae

Prior to this study, three plastomes of *Scutellaria* were available on GenBank, but two of them were without any related publication or analysis; only *S. baicalensis* was formally published [50]. The species *S. indica* var. *coccinea* has since been published, but the sequences were not

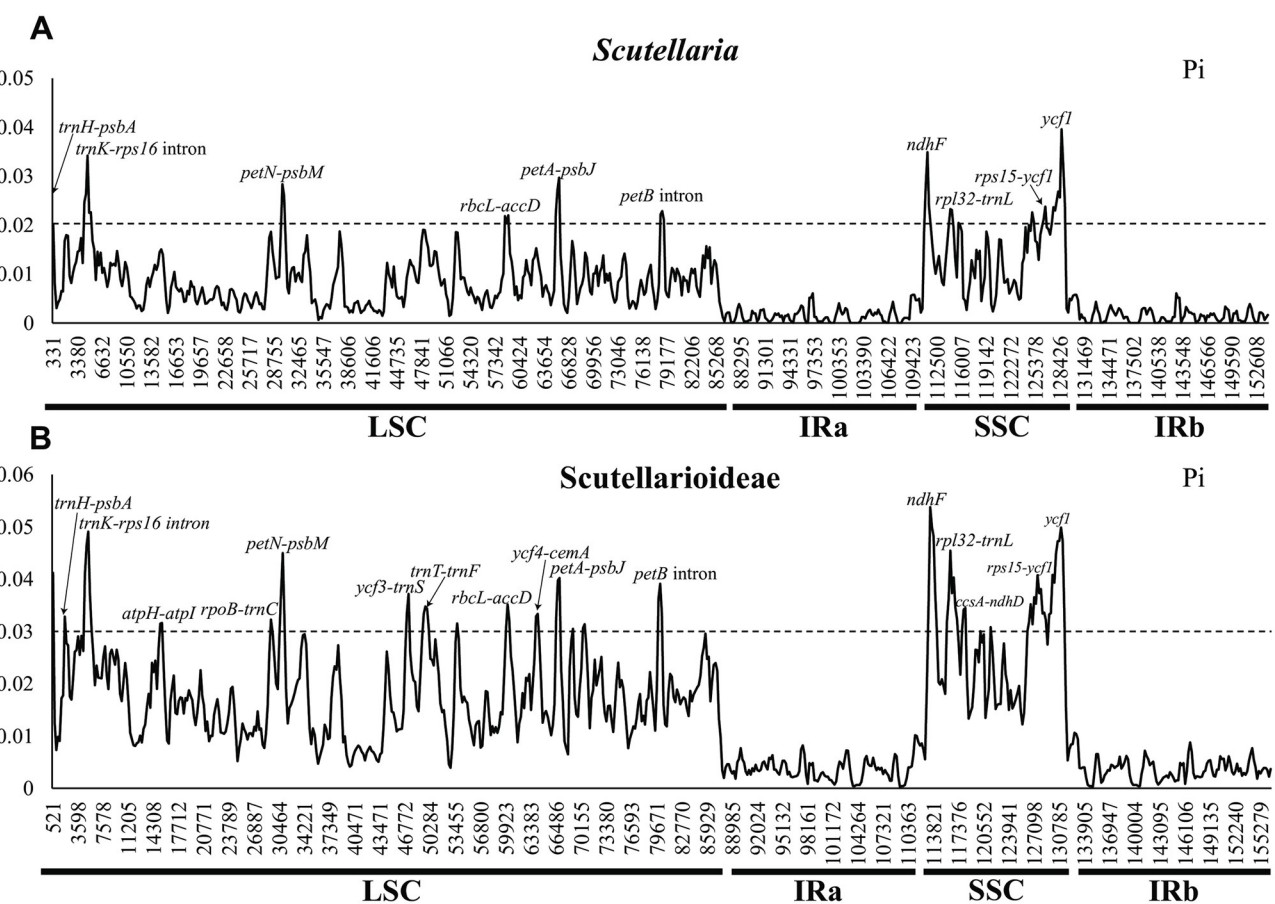

**Fig 7. Sliding window analysis of the whole chloroplast genomes. (A)** the 11 species of *Scutellaria*; **(B)** the 15 samples of Scutellarioideae.

yet available [51]. Here, we report on 12 complete plastomes representing 11 species from four genera of Scutellarioideae for the first time. In total, 15 plastomes were included for comparative analysis.

The length of plastomes of the 15 taxa from Scutellarioideae ranged from 151,675 bp to 153,272 bp, with the variation mainly caused by large indels (insertions/deletions) in the non-coding regions. The plastomes of Scutellarioideae are highly conserved in structure, gene

**Table 4. The number of parsimony-informative sites and the best fit model for each data set.**

| Data set* | Aligned length [bp] | GC content (%) | No. of variable sites [bp] | No. of parsimony-informative sites [bp] | Best fit model (BIC) |
|---|---|---|---|---|---|
| CR | 70,046 | 38.20 | 14,288 (20.4%) | 8,695 (12.41%) | GTR+I+Γ |
| NCR | 72,624 | 33.60 | 23,032 (31.71%) | 13,208 (18.19%) | GTR+I+Γ |
| CPG | 144,120 | 37.10 | 36,934 (25.63%) | 21,763 (15.10%) | GTR+I+Γ |
| 16VAR | 24,090 | 31.60 | 9,953 (39.8%) | 5,865 (24.30%) | GTR+I+Γ |
| 6CP | 8,346 | 35.70 | 2,820 (33.6%) | 1,812 (21.71%) | GTR+I+Γ |

*: CPG, complete plastome sequences; CR, coding regions; NCR, non-coding regions; 16VAR: 16 hyper-variable regions; 6CP: six commonly cpDNA regions.

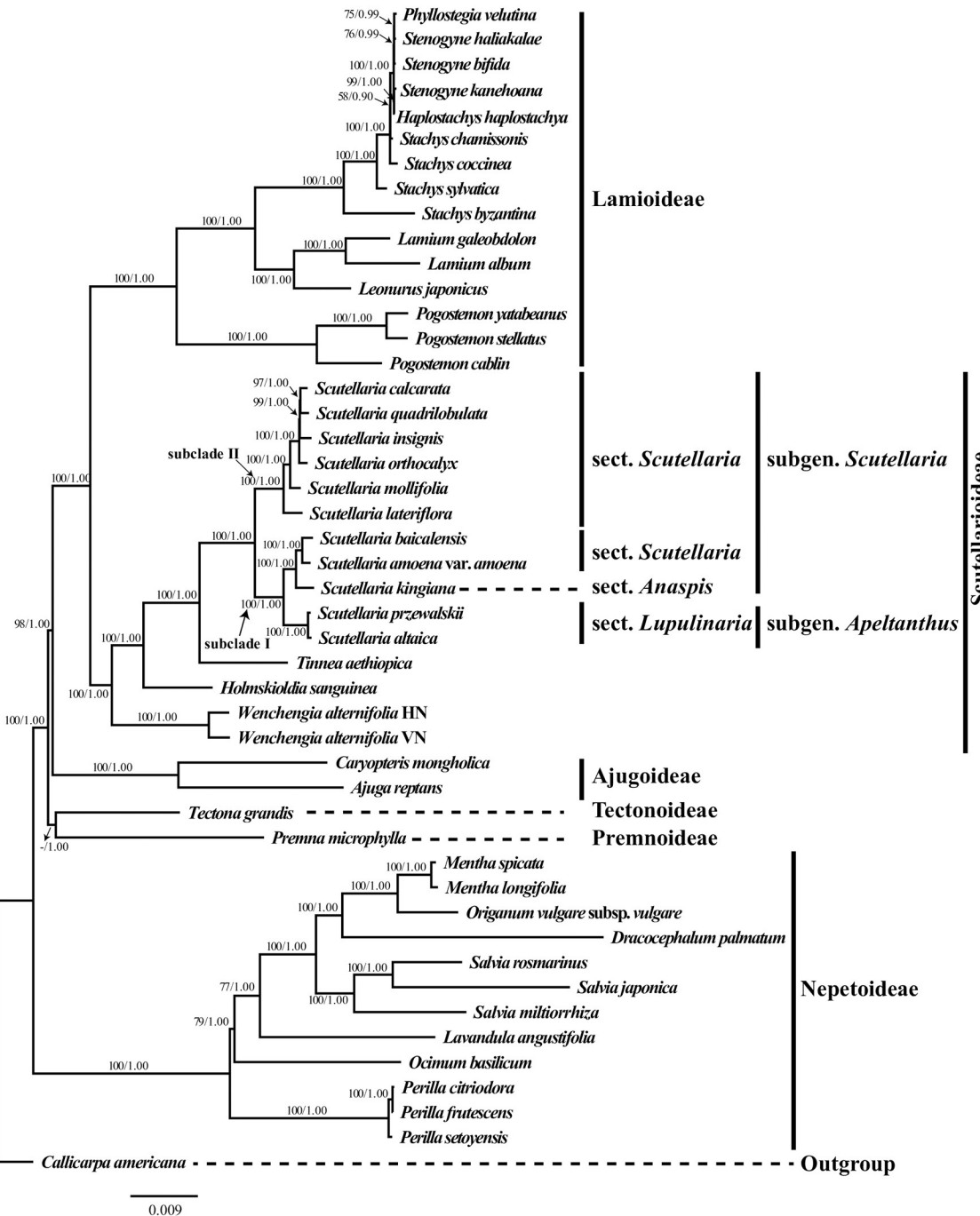

**Fig 8. The best-score tree from maximum likelihood analysis of Scutellarioideae based on the complete plastome sequences.**
Support values BS ≥ 50% or PP ≥ 0.90 are displayed on the branches follow the order ML$_{BS}$/BI$_{PP}$ ("-" indicates a support value BS < 50%). Scale bar denotes the expected number of substitutions per site in maximum likelihood analysis.

order, and content. All the 15 plastomes encode 114 unique genes in the same gene order and display the typical quadripartite structure, including a pair of IR regions separated by the LSC and SSC regions (Fig 1 and S1 Fig). Lee and Kim [51] have recently identified 115 genes from the plastome of *S. indica* var. *coccinea*. In comparison with the present study, one extra tRNA

gene was identified. Because sequences and annotation information of this plastome have not been released, we could not include it for comparative analysis. The average GC content of Scutellarioideae plastomes in our study was 38.3%, very similar to other species in Lamiaceae [50, 51, 75–77].

The complete aligned sequences indicate that the 15 plastomes of Scutellarioideae are conserved, with the sequence identity among genera higher than 95% and no major structural rearrangements or gene losses discovered. The location of the IR boundaries, especially as this pertains to IR contraction and expansion, can be exploited for phylogenetic purposes as small expansions or contractions tend to have similar endpoints in closely related species [78]. We find that the variation in the IR boundaries in Scutellarioideae, however, is not as extensive as reported in previous studies [79].

Chen et al. [79] reported that the LSC/IR regions within Lamiales can be divided into four different types: type I, with the LSC/IR regions being located in the intergenic *rpl2-rps19*; type II, with the *rps19* pseudogene at the LSC/IR border; type III, with the *ycf2* pseudogene at the IR/LSC border; and type IV, with the IR extending to include the *trnH* gene and a truncated *psbA* pseudogene at the IR/LSC border. Subsequently, Gao et al. [48] detected a new type where the IR/LSC border was found in the intergenic *rpl2-rps19*. In our study, the LSC/IR junction of all 15 species of Scutellarioideae belongs to type II, and the boundary of the SSC and IRa regions in *Wenchengia alternifolia* and *Tinnea aethiopica* is aberrant, with an expansion that involved the complete *ndhF* gene being included in the SSC region (Fig 6).

SSRs are widely used in molecular identification, genetic diversity, and population genetics studies [80]. Studies have shown that A/T mononucleotides are often very rich in SSRs [50, 76, 77]. Our analyses also show that SSRs in Scutellarioideae are generally composed of short polyadenine (poly A) or polythymine (poly T) repeats and rarely contain tandem guanine (G) and/ or cytosine (C). In this study, a total of 455 SSRs are made up of A or T bases, accounting for approximately 77% of the total SSRs. In addition, most mononucleotide repeats were detected in the non-coding regions (S3 Table). A potential reason for the higher frequencies of the AT repeats is the strand separation for ATs is relatively easier than GCs during plastome replication, which increases slipped-strand mispairing. There is a tendency for SSRs to occur in the non-coding region of the chloroplast genome of higher plants [81]. The molecular processes that give rise to repeats are more likely to be preserved in non-coding regions because there is strong selection against them in coding regions. In addition, because the non-coding regions are so AT rich, there is an expectation that repeats will be biased towards AT content, especially in the single copy regions. In general, the structure and organization of plastomes is conserved and SSRs primers are transferable across species or genera. Thus, the new SSRs detected in this study are potential resources for estimating the genetic diversity of some important medicinal species of *Scutellaria*, and for phylogenetic study among species and genera.

It has been demonstrated that short dispersed repeats are a major factor promoting plastome rearrangements in land plants [82], but within the unrearranged plastid sequence the function of these repeats remains unknown [76]. Our study reveals three types of repeats (forward, reverse, and palindromic) in the 15 plastomes of Scutellarioideae. As has been reported in other species of Lamiales [79, 83], most of these repeats are located in the intergenic spacers and introns, but several also occur in the coding regions. In total, 22.04% of the repeats occur in four protein coding regions (*psaB*, *psaA*, *ycf1*, and *ycf2*; S4 Table). The genes *ycf1* and *ycf2* have been demonstrated to be associated with repeat events [84]. In our study, the richest repeats are found in the *ycf2* gene, similar to other studies [48, 79, 83]. However, only one palindromic repeat, in the *ycf1* gene of *Wenchengia alternifolia* VN was detected. The absence of the dispersed repeats from the *ycf1* gene in this study is partially because the plastomes from closely related species are highly similar and lack of variation.

## Potential DNA barcodes for *Scutellaria*

Genomic comparative analyses of complete plastome sequences have become necessary for developing variable DNA barcodes, especially for finding mutation "hotspot" regions for novel DNA barcodes in addition to the set of widely used DNA markers (*matK*, *rbcL*, *psbA-trnH*, and nrITS [85–87]).

Though *Scutellaria* is the second largest genus within Lamiaceae and has medicinally important [88], DNA barcoding research within the genus is wanting. Guo et al. [68] attempted to distinguish the most widely used medicinal species, *S. baicalensis*, from its congeners, *S. amoena*, *S. rehderiana* Diels, and *S. viscidula* Bunge. However, this study had sparse sampling and only three DNA regions were used (*matK*, *rbcL*, and *psbA-trnH*). In previous studies, the cpDNA markers *rps16* (as part of the *trnK-rps16* intron), *ndhF*, *rps15-ycf1*, and *ycf1* were used to resolve the systematic position of some genera within Lamiaceae [89, 90], and fragments of *psbA-trnH*, *rpl32-trnL*, *rps15-ycf1*, and *ycf1* were applied to infer the intrageneric relationships [91, 92]. Some fragments, such as *petN-psbM* and *petA-psbJ* have been commonly used in seed plant phylogenetic studies [93, 94], but never have been used to resolve phylogenetic relationships in Lamiaceae. The intergenic spacer *rbcL-accD* and *petB-petD* intron have been identified as highly variable regions in other plants [95, 96]. The 10 highly variable regions (*psbA-trnH*, *trnK-rps16* intron, *petN-psbM*, *rbcL-accD*, *petA-psbJ*, *petB-petD* intron, *ndhF*, *rpl32-trnL*, *rps15-ycf1*, and *ycf1*; Fig 7) identified here could be used as potential barcodes for species identification and phylogenetic study of *Scutellaria*. Although further research is needed to investigate the reliability and effectiveness of using these regions and/or complete plastome sequences for DNA barcodes in *Scutellaria*, the results obtained here could be a reference for future studies on global genetic diversity assessment, phylogeny, and population genetics.

## Phylogenetic relationships within Scutellarioideae

Our study is the first to use complete plastome sequences to reconstruct the phylogeny of Scutellarioideae. The phylogenetic tree obtained here is largely consistent with previous studies based on the plastid DNA markers [1, 9, 97, 98]. However, some phylogenetic relationships within Lamiaceae differ from recent nuclear trees [99]. Such incongruence between plastid and nuclear phylogenies emphasizes a need for phylogenetic inferences based on both plastome sequences and nuclear data, which can together both robustly resolve relationships and point to potential ancient hybridization events.

The monophyly of Scutellarioideae is confirmed based on the analyses of all datasets (Fig 8, S3–S8 Figs), and the major splits determined in this study for Scutellarioideae agree with previous studies [1, 9]. This study confirmed that the monotypic genus *Wenchengia* is sister to the remainder of Scutellarioideae (Fig 8). This relationship has been reported in a previous study using two DNA markers (i.e. *rbcL* and *ndhF*; [9]). The accession of *W. alternifolia* from Vietnam was recovered in a clade with an accession of *W. alternifolia* from Hainan, China in our analyses. The genus has long been thought to be endemic to Hainan Island in China and was only recently reported from Vietnam. As suggested by Paton et al. [16], the distribution of *Wenchengia* in Vietnam indicates that the Hainan populations are probably relicts of a once more widely distributed *W. alternifolia*. The discovery of living plants in Vietnam offers the opportunity for population genetic and biogeographic studies of *Wenchengia* in future.

The African genus *Tinnea* is sister to *Scutellaria*, as reported by Wagstaff et al. [8] and Li et al. [1, 9]. Although *Renschia* has never been included in a molecular analysis, morphological characters, e.g. ciliate anthers, well-developed nectar disk, bilabiate calyx with entire, rounded lips, and the closing of the calyx during fruit maturation [6]), suggest a close relationship

among *Renschia*, *Tinnea*, and *Scutellaria*. *Renschia* is probably most closely related to *Tinnea* based on distribution (both genera are distributed in Africa; *Renschia* is endemic to North Somalia and *Tinnea* to tropical Africa) and morphology. Vatke [100] established *Renschia* based on *Tinnea heterotypica* S. Moore, and distinguished *Renschia* from *Tinnea* by its protruding stamens, the short and basal areoles of nutlets, and the indistinct nervation of calyces.

A total of 11 species of *Scutellaria* were sampled from both subgenera sensu Paton [5]. The monophyly of *Scutellaria* is supported here as in other studies [1, 9, 18, 43], but the infrageneric classification of *Scutellaria* as proposed by Paton [5] is not supported by the present study (Fig 8). As shown in Fig 8, in our sampling *Scutellaria* is comprised of two subclades: Subclade I included five taxa from subg. *Scutellaria* and two taxa from subg. *Apeltanthus*; Subclade II consists of six species from subg. *Scutellaria* sect. *Scutellaria*. Species from sect. *Scutellaria* are recovered in both subclades, thus the monophyly of subgenus *Scutellaria* and sect. *Scutellaria* is not supported by the plastome sequences in this study or nuclear ribosomal sequences in previous studies [18, 43]. With only one species of sect. *Anaspis* sampled here, it is premature to assess its monophyly. Though a recent study by Safikhani et al. [18] revealed that sect. *Anaspis* is a well-supported group, only four representatives of the section from Iran were included in their study. Subgenus *Apeltanthus* is well supported in all studies [18, 43]. The two sections of subg. *Apeltanthus*, sect. *Apeltanthus* and sect. *Lupulinaria*, are shown to be monophyletic in our study as in Zhao et al. [43]. However, based on a broader sampling, Safikhani et al. [18] revealed that neither of the two sections is supported. Further phylogenetic study of subg. *Apeltanthus* is needed based on a more comprehensive sampling and more DNA markers.

Despite the limited sampling, our study, based on complete plastomes, presents a more resolved and better supported phylogeny of Scutellarioideae than previous studies [1, 9, 18, 43, 98]. All the phylogenetic trees inferred from the complete plastome sequences have higher resolution (Fig 8) than trees based on the six commonly used chloroplast DNA regions (*matK*, *ndhF*, *rbcL*, *rpL32-trnL*, *rps16-intron*, and *trnL-F*; S7 Fig) in previous studies [9, 41, 68] and 16 hyper-variable chloroplast regions (S8 Fig), demonstrating that complete plastome sequences can markedly improve phylogenetic resolution, at least within Scutellarioideae and Lamiaceae.

## Supporting information

**S1 Table. Complete chloroplast genome samples to the Scutellarioideae phylogenetic analysis.**
(XLSX)

**S2 Table. The proportion of protein-coding length, tRNA length, and rRNA length in total sequence.**
(XLSX)

**S3 Table. Statistics of simple sequence repeats in each species of Scutellarioideae.**
(XLSX)

**S4 Table. Statistics of longer repeats in each species of Scutellarioideae.**
(XLSX)

**S1 Fig. Gene map of the complete chloroplast genome of Scutellarioideae.**
(PDF)

**S2 Fig. Progressive Mauve alignment among the species of Scutellarioideae.**
(PDF)

**S3 Fig. Maximum parsimony majority-rule consensus tree of Scutellarioideae resulting from coding regions (CR) dataset.** Bootstrap values > 50% are indicated at individual branches.
(PDF)

**S4 Fig. The Bayesian 50% majority-rule consensus tree of Scutellarioideae based on coding regions (CR) dataset.** Bayesian posterior probabilities ≥ 0.95 are indicated at individual branches.
(PDF)

**S5 Fig. Maximum parsimony majority-rule consensus tree of Scutellarioideae resulting from non-coding regions (NCR) dataset.** Bootstrap values > 50% are indicated at individual branches.
(PDF)

**S6 Fig. The Bayesian 50% majority-rule consensus tree of Scutellarioideae based on non-coding regions (NCR) dataset.** Bayesian posterior probabilities ≥ 0.95 are indicated at individual branches.
(PDF)

**S7 Fig. The best-score tree from maximum likelihood analysis of Scutellarioideae based on the combined dataset the most commonly used DNA markers (*matK*, *ndhF*, *rbcL*, *rpL32-trnL*, *rps16-intron* and *trnL-F*) in the previous studies.** Support values BS ≥ 50% or PP ≥ 0.90 are displayed on the branches follow the order $ML_{BS}/BI_{PP}$ ("-" indicates a support value BS < 50%). Scale bar denotes the expected number of substitutions per site in maximum likelihood analysis.
(PDF)

**S8 Fig. The best-score tree from maximum likelihood analysis of Scutellarioideae based on the combined dataset of thesixteen hyper-variable regions.** Support values BS ≥ 50% or PP ≥ 0.90 are displayed on the branches follow the order $ML_{BS}/BI_{PP}$ ("-" indicates a support value BS < 50%). Scale bar denotes the expected number of substitutions per site in maximum likelihood analysis.
(PDF)

## Acknowledgments

The authors are grateful to Prof. Shi-Xiao Luo, Prof. Shen-Zhuo Huang, Mr. Hong-Liang Chen, Mr. Yi Yang, Miss Yuan-Yuan Li, and Miss Qiao-Rong Zhang for their assistance in sample collection. We also thank Dr. Richard Olmstead and another anonymous reviewer for their constructive suggestions that greatly improved the paper.

## Author Contributions

**Conceptualization:** Fei Zhao, Hua Peng, Chun-Lei Xiang.

**Data curation:** Fei Zhao, Bo Li, Ya-Ping Chen, Wen-Bin Yu, En-De Liu.

**Formal analysis:** Fei Zhao, Bo Li, Bryan T. Drew, Ya-Ping Chen, Wen-Bin Yu.

**Funding acquisition:** Qiang Wang, Chun-Lei Xiang.

**Resources:** Qiang Wang, Wen-Bin Yu, En-De Liu, Yasaman Salmaki, Chun-Lei Xiang.

**Visualization:** Chun-Lei Xiang.

**Writing – original draft:** Fei Zhao, Bo Li, Chun-Lei Xiang.

**Writing – review & editing:** Fei Zhao, Bo Li, Bryan T. Drew, Ya-Ping Chen, Qiang Wang, Wen-Bin Yu, Yasaman Salmaki, Hua Peng, Chun-Lei Xiang.

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
