## [Decision Letter · Decision Letter 0]

20 Dec 2019

PONE-D-19-30248

Leveraging plastomes for comparative analysis and phylogenomic inference within Scutellarioideae (Lamiaceae)

PLOS ONE

Dear Dr. Xiang,

Thank you for submitting your manuscript to PLOS ONE. After careful consideration, we feel that it has merit but does not fully meet PLOS ONE’s publication criteria as it currently stands. Therefore, we invite you to submit a revised version of the manuscript that addresses the points raised during the review process.

We would appreciate receiving your revised manuscript by Feb 03 2020 11:59PM. To enhance the reproducibility of your results, we recommend that if applicable you deposit your laboratory protocols in protocols.io, where a protocol can be assigned its own identifier (DOI) such that it can be cited independently in the future. For instructions see: http://journals.plos.org/plosone/s/submission-guidelines#loc-laboratory-protocols

We look forward to receiving your revised manuscript.

Kind regards,

Genlou Sun

Academic Editor

PLOS ONE

Journal Requirements:

2. In your Methods section, please provide additional information regarding the permits you obtained for the collection of plant materials. Please ensure you have included the full name of the authority that approved the field site access and, if no permits were required, a brief statement explaining why.

Reviewers' comments:

Reviewer's Responses to Questions

**Comments to the Author**

1. Is the manuscript technically sound, and do the data support the conclusions?

Reviewer #1: Partly

Reviewer #2: Yes

2. Has the statistical analysis been performed appropriately and rigorously? 

Reviewer #1: Yes

Reviewer #2: Yes

3. Have the authors made all data underlying the findings in their manuscript fully available?

Reviewer #1: No

Reviewer #2: Yes

4. Is the manuscript presented in an intelligible fashion and written in standard English?

Reviewer #1: Yes

Reviewer #2: Yes

5. Review Comments to the Author

Reviewer #1: The manuscript by Fei Zhao et al. describes 12 new chloroplast genomes from 11 species and four genera of Scutellarioideae (Lamiaceae). The authors present a careful and thoughtful set of analyses and interpretations, characterizing genome structure and genic content, as well as genomic features and regions with utility for population genetic and phylogenomic studies of the clade. Regarding the latter, they attempt to demonstrate the utility of plastid phylogenomic data for resolving species-level relationships in Scutellarioideae, as well as within Lamiaceae.

I genuinely appreciate the authors’ approach to this study and have no major issues with their methods; most of my comments are intended as helpful suggestions. The pace at which new chloroplast genomes have been generated for Lamiaceae has been relatively slow, and structural variations within the family and the utility of large-scale plastid data in phylogenomic and population genetic studies of mints is not well established. Currently there are only a few published studies that have made these comparisons, and all are clade-specific. Thus, this study represents a new and important contribution to the literature.

I have only a few major concerns that I feel should be addressed by the authors:

1. (L196) The authors made a new chloroplast genome assembly for Callicarpa americana using data deposited in SRA by the Mint Evolutionary Genomics Consortium (2018), but they do not provide any details about this assembly in the manuscript or supplemental tables and have not included an accession ID indicating that the annotated genome assembly is available in an appropriate repository (or at least I am not seeing these details).

2. It’s not clear from the study results whether complete chloroplast sequence data are necessary to reconstruct relationships in Scutellarioideae or if a smaller matrix of highly variable chloroplast loci (e.g., the list characterized by this study) would also resolve species-level relationships with high support values. This is especially relevant given that sequence capture approaches have very recently become more efficient, reducing off-target reads from the plastome that can be mined and assembled for phylogenomic analyses from targeted capture and high-throughput sequencing of nuclear loci (see de La Harpe et al. 2018); additional financial investments for captures and/or sequencing is now necessary to ensure acquisition of large-scale plastid data for phylogenomic analyses. It is difficult to assess from the data presented here how much plastid data are needed to robustly resolve relationships, especially because comparisons of plastid phylogenomic results are made with published results based on different taxon and DNA sampling schemes. I think this “problem” could be addressed with minimal effort by the authors (see my comment/recommendation for L509 below).

Additional comments:

L27: It seems rather subjective to refer to Scutellaria as "one of the largest and most taxonomically challenging genera". Please explain.

L29: This is a picky (semantic) point. A “lack of molecular data” doesn’t directly hinder our understanding of phylogeny. It hinders our ability to accurately and robustly reconstruct phylogenetic relationships, which in turn hampers our understanding of diversity and evolutionary history.

L30-35. The authors should consider alternative phrasing here for improved efficiency.

L50. Pick one: “angiosperm family” or “flowering plants”.

L51. Why is Scutellarioideae one of the most distinctive subfamilies? Explain.

L97. The authors indirectly refer to molecular systematic studies of particular genera, but do not provide citations for these studies as examples. I think it’s fair to do so.

L105. The authors extensively discuss the need for broad or comprehensive sampling in Scutellaria/Scutellarioideae, and yet this study does not accommodate this need.

L159: Please note the correct spelling of “de Bruijn”. What criteria were used in the manual selection of sequence paths (via visualization in Bandage)? This information might be useful to readers.

L160: Were mapped reads also used to evaluate and report on depth of coverage?

L164: DOGMA is an “somewhat dated” tool for chloroplast genome annotation and, although that doesn’t mean it isn’t useful, I’m wondering why the authors chose this tool over newer options. Have they compared their DOGMA annotations with GeSeq (Tillich et al. 2017) annotations?

L191–192. This wording isn’t clear. Do you mean that you “evaluated the utility of complete plastome sequences for resolving both the placement of Scutellarioideae in Lamiaceae and species-level relationships in the clade (= traditional subfamily)”?

The author’s use of Lamiaceae-wide sampling seems to go beyond the explicitly stated goals of this study, which focus on Scutellarioideae/Scutellaria (especially without the wording I mention above). Since there isn’t a backbone phylogeny based for Lamiaceae inferred from complete plastome sequences, they may be underselling the significance of their plastid phylogenomic results and its value for the Lamiaceae systematics community. Lamiaceae-wide relationships are only briefly mentioned in the discussion and the importance of the placement of Scutellarioideae in Lamiaceae might be worth highlighting.

L205–206. Which IR copy was used (IRa or IRb)? If IRb was not used, how did you deal with the ycf1 gene that spans the IRb/SSC boundary? And did you remove the rps19 pseudogene?

L244. Previous reports by Jiang et al. (2017) and Lee and Kim (2019) report 114 and 115 genes in chloroplast genomes from Scutellaria sp., respectively. I don’t see any mention of the recent Lee and Kim (2019) manuscript on the S. indica var. coccinea chloroplast genome here. What is the additional (tRNA?) gene? Has it been missed in previous annotations and those included here? Given that new genomes are available since you started your study, it might be worth a quick look and reference in your discussion.

L405. What do the authors mean by “intense” here?

L409–411. I appreciate the discussion of types I–IV here. This is interesting. It would be useful to mention any lineage-specific patterns with regard to types observed in Lamiales (if known by the authors).

L444. “Though Scutellaria is one of the largest and most important medicinal genera within Lamiaceae”. Is there an appropriate citation for this? This statement seems subjective.

L467. The phylogeny is consistent with studies based primarily on plastid data. However, your results differ from recent nuclear trees (i.e. Mint Evolutionary Genomics Consortium 2018). The incongruence between plastid and nuclear phylogenies emphasizes a need for phylogenetic inferences based on complete plastome sequences, which can robustly resolve relationships (as the authors attempt to demonstrate in this study). Perhaps that is worth mentioning in your discussion.

L509. It is not unreasonable to expect strongly supported relationships in a phylogenetic study with sparse taxon sampling, regardless of whether genome-scale DNA sequence data were used. The authors could easily bolster their argument that use of complete plastome data improves phylogenetic resolution/support if they showed a side-by-side comparison of phylogenetic results, e.g.: (1) phylogeny inferred from commonly used or highly variable chloroplast markers (mentioned in the manuscript) subsampled from their matrix, and (2) your Figure 7 tree inferred from whole chloroplast sequences. This would demonstrate the utility of large-scale data and help avoid making comparisons of phylogenies yielded from studies with different taxon/DNA sampling schemes.

Reviewer #2: Title: Leveraging plastomes for comparative analysis and phylogenomic inference within Scutellarioideae (Lamiaceae).

Authors: F. Zhao et al.

Journal: PLOS One

Review:

This manuscript describes the comparison of 12 new plastid genomes along with 3 previously released plastomes for a total of 15 plastomes of the subfamily Scutellarioideae. The clade consists of the large genus Scutellaria, with 300-400 species and four small genera (three monotypic). Plastome structure, sequence variation, and repeats are presented, along with a phylogenetic tree based on whole genome sequence data.

The presentation is straightforward and mostly descriptive. I have a few more important points, but most of my comments are minor and, I hope, will make the paper a little clearer in places.

Major points.

1. GC content and distribution of SSR and repeats. It has been known for 30 years or more that chloroplast genomes are AT rich, especially in the non-coding regions, where selection is not maintaining GC content for amino acid coding. This paper notes the differences in GC content between large and small single copy regions and the inverted repeats, but doesn’t mention the difference between coding/non-coding regions (Table 4 has GC content for the coding and non-coding datasets, but these are not for complete genomes, if I understand correctly). Separately, the authors note that most of the mono- and dinucleotide repeats are ATs and that they are located primarily in the non-coding regions, but they never put these observations together to conclude that the molecular processes that give rise to repeats (errors in replication) are most likely to be preserved in non-coding regions, because there is strong selection against them in coding regions, and that because the non-coding regions are so AT rich there is an expectation that repeats will be biased towards AT content, especially in the single copy regions. I think it would be great to note this association in the Discussion.

2. Counts of SSR and repeat regions in the plastomes are given for individual plastomes and summed for all plastomes (depicted in Figs 2-3), but the aggregate numbers are not very meaningful, because in many, perhaps most, cases these are shared among genomes by descent from a common ancestor. These plastomes represent closely related species that have diverged little in primary sequence, as they show elsewhere in this paper, so the expectation would be that many of these features are shared by descent. This information would be of great interest to people interested in plastome evolution and in the utility of SSR data for evolutionary studies. I would like to see some analysis of the extent to which these repeat elements are shared between related species.

3. The graphic depictions of the repeat data in Figs. 2 and 3 don’t work very well for me. In my opinion, for example, the Table S2 depicts these data better and more precisely than the bar graphs and pie charts in Fig. 2. This is especially so in fig. 2D, where the many colors are not easily distinguished and the bars are so compressed for all but the A/T mononucleotide repeats that they cannot be interpreted. My point above about repeats shared by common ancestry is important here. By summing the SSRs or repeats in bar graphs like this, the reader is led to believe that there are many independent repeats, when there are likely to be only a few that are shared among many or all sampled species. It would be really interesting to know how many are shared and how widely among Scutellarioideae and how many are unique to individual plastomes.

4. I’m a little confused about a few of the Simple Sequence Repeats in Fig. 2D. My understanding of SSRs is incomplete, but I understand that they are repeat motifs of one or a few nucleotides, which exist, of course, in their complement on the opposite strand. Hence repeats like AACC and GGTT count as one repeat (AACC/GGTT in Fig. 2D), because they will exist in exactly equal numbers as complements on opposing strands of DNA and are for all intents and purposes indistinguishable when assessing DNA variation. However, three of the SSRs presented in Fig. 2D and in the text on page 14 are not complement pairs, so I don’t know if I misunderstand their interpretation of SSRs or if there is an error in their results. Specifically, they include the following: AATC/ATTG, AAATC/ATTTG, and AAATAG/ATTTCT.

5. IR/SSR boundary presentation. In Fig. 5 and Results, page 16, for Holmskioldia and all of the Scutellaria accessions, the SSC/IRa junction is depicted as falling in the middle of ndhF, while the SSC/IRb junction is depicted as falling in the middle of ycf1. There is something wrong here. Since, by definition the two IR regions are identical, inverted sequence regions, the IR sequence adjoining the two SSC junctions have to be identical. The IR sequence can either be ndhF OR ycf1, but not ndhF on IRa and ycf1 on IRb. If the SSC/IRa junction has moved 25-40 bp into ndhF, then these nucleotides will also appear in the IRb at its SSC junction. If the ycf1 coding frame extends across the SSC/IRb junction, then it would be disrupted by including this sequence in the middle of its reading frame. It is possible that this is the case and that ycf1 is a non-functional pseudogene in Scutellaria (as it is in a number of other plastomes). I think the authors need to look closely at those junctions and sort this out.

In Fig. 5, I think it would be good to show the portion of the ycf1 gene that exists in the SSC end of the IRa, as is done with the rps19 fragment in the IRb at the LSC/IRb junction.

Minor comments in order of appearance:

INTRODUCTION

a. P. 2, line 49. Introduction misspelled

b. P. 3, line 61. Two early molecular phylogenetic studies that confirmed Cantino’s evidence for the ‘modern’ circumscription I suggest that the one or both of these papers should be cited here to acknowledge that fact. See also p. 22, lines 467.

Wagstaff, S. J. and R. G. Olmstead. 1997. Phylogeny of the Labiatae and Verbenaceae inferred from rbcL sequences. Syst. Bot. 22: 165-179.

Wagstaff, S. J., P. A. Reeves, L. Hickerson, R. E. Spangler, and R. G. Olmstead. 1998. Phylogeny of Labiatae s.l. inferred from cpDNA sequences. Pl. Syst. Evol. 209: 265-274.

c. P. 5, line 113. Plastomes are described as “…multiple copies, and a typically quadripartite…” Add “per cell” after “multiple copies.”

d. METHODS

e. P. 8, lines 175-6. What is “hamming distance?” What was the basis for a minimum repeat size of 30?

f. RESULTS

g. P. 10 lines 221-2. Describing coverage to 3 decimal places, when coverage varies throughout the plastome seems overly precise. Close enough to take it to the nearest whole intger.

h. P. 10, lines 226-7. “The GC content was evenly distributed …” What does this mean? Clearly it is not evenly distributed throughout the genome. I think the authors mean that all of the plastomes in Scutellarioideae have similar overall GC content.

i. P. 10, lines 228-9. Please also note the difference between coding and non-coding regions in GC content.

j. P. 11, Fig. 1. Figure 5 shows the gene rps19 at the LSC/IRa junction, but this gene is not depicted on any of the circular diagrams in Fig. 1. Is anything else missing?

Also, gene fragments that exist on one IR, because they are part of a gene split by the opposite SC/IR junction (e.g., ycf1) should be identified as pseudogenes with an appropriate symbol.

k. P. 11, lines 245-6. “A total of eighteen genes have undergone duplication in the IR region” This should read: “A total of eighteen genes exist in duplicate copies in the IR region.” There is no reason to believe that these have undergone duplication, since the IR is a feature of virtually all plastomes.

l. P. 14, line 275. “The length of repeats ranged from 10 to 139, with an average value of 17 bp.” This is not very meaningful, since a repeat length of 10 was arbitrarily chosen as the lower cutoff for this analysis. Also, I don’t believe the average repeat length is very meaningful, given the fact that many of the repeats will be present in multiple plastomes due to common descent.

m. P. 15, line 294. I would like examples or descriptions of what is meant by forward, reverse, and palindromic repeats.

n. P. 16. Lines 325-31. Figure 4. The caption and the legend on the figure need to be aligned better. Does “protein coding” = “exon,” “intron” = “UTR,” etc. I am confused about what is meant by mRNA. mRNA molecules are the product of transcription of protein coding genes and are not, themselves part of a genome. There doesn’t appear to be any gray portions of these plastome figures; does mRNA refer to the arrows above the linear genomes? Why the very long arrow from ca. 99k to ca. 70k labeled rps12? Isn’t this a trans-spliced mRNA and not one very long transcription unit?

o. P. 16, lines 332-3. See point #5 above regarding ndhF and ycf1 genes at the SSC/IR junctions.

p. P. 18, line 375. Caption to Fig. 7. Move the greater-than-or-equal-to sign to be in front of 50%.

q. DISCUSSION

r. P. 19, lines 402-3. “…IR contraction and expansion, has been considered to be a factor underlying species evolution withibn land plants.” I am unaware of any suggestions that the expansion/contraction of the IR in plastomes has any functional role that may impact speciation or evolution in any way. The cited papers do not mention anything like this. I think the authors mean to say that the variation in IR junction can have phylogenetic signal among relatively closely related species.

s. P. 19, line 405. “Extensive” instead of “intense”

t. P. 20, lines 428-9. “short dispersed repeats are a major factor promoting plastome rearrangements in land plants, the function of these repeats remains unknown.” Repeat regions have been identified at the end points of inversion in plastomes, but I think most people think this is a random process and that selection acts against most such mutations if they disrupt coding regions or regulatory genomic elements.

u. P. 20, lines 437-8. “The absence of dispersed repeats from the ycf1 gene in this study may partially explain the sequence conservation of plastomes of Scutellarioideae.” These plastomes are highly similar sequences (not necessarily “conserved”), because they are from closely related species. There may be fewer repeats in ycf1 than in plastomes of some other groups, but I doubt that it is a causative agent for the lack of variation in these plastomes.

v. P. 23, line 494. “As shown in Fig. 7, Scutellaria is comprised of two subclades…” Add “in our sampling” to this sentence in front of “Scutellaria.” The sampling is so limited, relative to the diversity of the genus, that this statement is a bit to strong.

w. P. 24, line 521. Fig. S1. What is the zig-zag redline that connects down across the plastomes in this figure?

Signed: Richard Olmstead

6. PLOS authors have the option to publish the peer review history of their article (what does this mean?). If published, this will include your full peer review and any attached files.

Reviewer #1: No

Reviewer #2: Yes: Richard G. Olmstead

---

## [Author Response · Author response to Decision Letter 0]

5 Feb 2020

Reviewer 1

The manuscript by Fei Zhao et al. describes 12 new chloroplast genomes from 11 species and four genera of Scutellarioideae (Lamiaceae). The authors present a careful and thoughtful set of analyses and interpretations, characterizing genome structure and genic content, as well as genomic features and regions with utility for population genetic and phylogenomic studies of the clade. Regarding the latter, they attempt to demonstrate the utility of plastid phylogenomic data for resolving species-level relationships in Scutellarioideae, as well as within Lamiaceae.

I genuinely appreciate the authors’ approach to this study and have no major issues with their methods; most of my comments are intended as helpful suggestions. The pace at which new chloroplast genomes have been generated for Lamiaceae has been relatively slow, and structural variations within the family and the utility of large-scale plastid data in phylogenomic and population genetic studies of mints is not well established. Currently there are only a few published studies that have made these comparisons, and all are clade-specific. Thus, this study represents a new and important contribution to the literature.

I have only a few major concerns that I feel should be addressed by the authors:

1. (L196) The authors made a new chloroplast genome assembly for Callicarpa americana using data deposited in SRA by the Mint Evolutionary Genomics Consortium (2018), but they do not provide any details about this assembly in the manuscript or supplemental tables and have not included an accession ID indicating that the annotated genome assembly is available in an appropriate repository (or at least I am not seeing these details).

Corrected. The complete chloroplast genomes of Callicarpa americana were reassembled based on the SRA data (SRR6940059), and thus we didn’t submit it to GenBank and only cited the SRA number in Table S4. Now, we have submitted the complete chloroplast genome of Callicarpa americana to GenBank; the accession number is MN883825.

2. It’s not clear from the study results whether complete chloroplast sequence data are necessary to reconstruct relationships in Scutellarioideae or if a smaller matrix of highly variable chloroplast loci (e.g., the list characterized by this study) would also resolve species-level relationships with high support values. This is especially relevant given that sequence capture approaches have very recently become more efficient, reducing off-target reads from the plastome that can be mined and assembled for phylogenomic analyses from targeted capture and high-throughput sequencing of nuclear loci (see de La Harpe et al. 2018); additional financial investments for captures and/or sequencing is now necessary to ensure acquisition of large-scale plastid data for phylogenomic analyses. It is difficult to assess from the data presented here how much plastid data are needed to robustly resolve relationships, especially because comparisons of plastid phylogenomic results are made with published results based on different taxon and DNA sampling schemes. I think this “problem” could be addressed with minimal effort by the authors (see my comment/recommendation for L509 below).

Corrected. Based on the results in the present study, we can say that using complete chloroplast sequence data or matrix of highly variable chloroplast loci are necessary to reconstruct relationships in Scutellarioideae. As suggested by the reviewer, we also selected some commonly used and highly variable chloroplast markers of all sampled taxa in the present study to reconstruct the phylogeny for comparison. Accordingly, more details are provided in the Discussion section. 

Additional comments:

L27: It seems rather subjective to refer to Scutellaria as "one of the largest and most taxonomically challenging genera". Please explain.

Corrected. We have rewritten this sentence to minimize the subjectivity (“Scutellaria is the second largest and one of the more taxonomically challenging genera within Lamiaceae…”)

L29: This is a picky (semantic) point. A “lack of molecular data” doesn’t directly hinder our understanding of phylogeny. It hinders our ability to accurately and robustly reconstruct phylogenetic relationships, which in turn hampers our understanding of diversity and evolutionary history.

We re-worded this section.

L30-35. The authors should consider alternative phrasing here for improved efficiency.

Corrected.

L50. Pick one: “angiosperm family” or “flowering plants”.

Corrected.

L51. Why is Scutellarioideae one of the most distinctive subfamilies? Explain.

The reason we say that is mostly based on its distinctive morphological characters: the entire-lipped, bilabiate calyx which closed at the mouth in fruit; the corolla zygomorphic, usually 2-lipped or 4-5-lobed; the anterior stamens have unilocular thecae due to the abortion of the upper locule, often referred to as dimidiate stamens; style terminal to sub-terminal, and the nutlets surface frequently tuberculate or bearing long, hair-like processes, attachment-scar usually lateral. The combination of these characters makes this subfamily easily distinguished from other subfamilies of Lamiaceae. 

We added “morphologically” to the sentence in question to clarify.

L97. The authors indirectly refer to molecular systematic studies of particular genera, but do not provide citations for these studies as examples. I think it’s fair to do so.

Corrected. we add the relevant reference in the review version. 

L105. The authors extensively discuss the need for broad or comprehensive sampling in Scutellaria/S

---

## [Decision Letter · Decision Letter 1]

6 Mar 2020

PONE-D-19-30248R1

Leveraging plastomes for comparative analysis and phylogenomic inference within Scutellarioideae (Lamiaceae)

PLOS ONE

Dear Dr. Xiang:

Thank you for submitting your manuscript to PLOS ONE. After careful consideration, we feel that it has merit but does not fully meet PLOS ONE’s publication criteria as it currently stands. Therefore, we invite you to submit a revised version of the manuscript that addresses the points raised duri

We would appreciate receiving your revised manuscript by Apr 20 2020 11:59PM. To enhance the reproducibility of your results, we recommend that if applicable you deposit your laboratory protocols in protocols.io, where a protocol can be assigned its own identifier (DOI) such that it can be cited independently in the future. For instructions see: http://journals.plos.org/plosone/s/submission-guidelines#loc-laboratory-protocols

We look forward to receiving your revised manuscript.

Kind regards,

Genlou Sun

Academic Editor

PLOS ONE

Reviewers' comments:

Reviewer's Responses to Questions

**Comments to the Author**

1. If the authors have adequately addressed your comments raised in a previous round of review and you feel that this manuscript is now acceptable for publication, you may indicate that here to bypass the “Comments to the Author” section, enter your conflict of interest statement in the “Confidential to Editor” section, and submit your "Accept" recommendation.

Reviewer #2: (No Response)

2. Is the manuscript technically sound, and do the data support the conclusions?

Reviewer #2: Yes

3. Has the statistical analysis been performed appropriately and rigorously? 

Reviewer #2: Yes

4. Have the authors made all data underlying the findings in their manuscript fully available?

Reviewer #2: Yes

5. Is the manuscript presented in an intelligible fashion and written in standard English?

Reviewer #2: Yes

6. Review Comments to the Author

Reviewer #2: Review of Second submission

Title: Leveraging plastomes for comparative analysis and phylogenomic inference within Scutellarioideae (Lamiaceae).

Authors: F. Zhao et al.

Journal: PLOS One

Review:

In this revision the manuscript is considerably improved and many of the criticisms I raised in my original review are addressed.

Major points:

1) Thanks to the authors for their reply to my query about the difference between forward, reverse, reverse complement, and palindromic repeats. It helped me to see the answer to a different concern I had raised regarding three of the SSRs that are not presented as complements in their paper. These are: AATC/ATTG, AAATC/ATTTG, and AAATAG/ATTTCT. I think these are labeled wrongly, with the mistake possibly being an improper computer output that was not caught by the authors, although I may still be mistaken in my understanding.

Let’s work an example.

If AATC/ATTG represents a reverse complement repeat, the reverse complement of AATC would be GATT.

In the genome the repeat would look like this: GATTGATTGATTGATTGATTGATTGATT etc.

I think an error worked its way into the designation of this SSR by shifting the reverse complement repeat by one nt from GATT to ATTG. Functionally, of course, there is no difference.

I think the proper way to present this in text and figures is as AATC/GATT, instead of AATC/ATTG, which does not make sense to me.

A similasr adjustment will realign the other two SSRs as well:

AAATC/ATTTG should be AAATC/GATTT (offset by one nt)

AAATAG/ATTTCT should be AAATAG/CTATTT (offset by two nt)

2) Figure 3 is a creative way to depict the numbers of different SSRs and how they are distributed among the taxa in the study. However, I don’t think the authors understood my concern about the shared ancestry of SSRs between related taxa. The new Fig. 3 shows how many copies of each identified SSR occur is each taxon’s plastome and these are summed across the circle in the individual SSR. However, this doesn’t indicate in any way whether an individual SSR is shared due to common ancestry or not. If the authors have mapped each SSR, they should be able to identify how many unique SSR loci there are. I suspect that there are some that might be shared by all and some that are unique to a single taxon. The number of unique loci will fall somewhere between 48, the maximum number found in one plastome, and 489, the total number of SSRs summed across all plastomes. For my money, this is the most interesting cross-taxon assessment of their data that they could do, but if it is a lot of work to do, I can understand their reluctance to include it.

3) I think there is still a problem with the depiction of the Inverted Repeat/Small Single Copy regions in Fig. 6. At the IRa/SSC boundary, in Holmskioldia and several spp. of Scutellaria, the end of the IRa that is closest to the SSC is shown having both a portio of ycf1 and ndhF. If the IR has migrated into the SSC in these taxa, then I think the ycf1 pseudogene should be offset from the boundary by at least the amount of ndhF that is now found in the IRa. They can’t both occupy the same space. I understand that these are not drawn to scale (e.g., in Holmskioldia there are 1077 nt of ycf1 and only 39 nt of ndhF), but the figure still depicts something that does not make sense. Similarly for those same taxa, the same amount of ndhF will be found in the IRb, but is not depicted there in fig. 6. If ycf1 crosses the IRb/SSC boundary, but the IR boundary has migrated into ndhF, so that a portion (39 bp and perhaps some flanking DNA) of ndhF is now in the IRb, then ycf1 must either be a pseudogene in that location, too, or the ndhF DNA is incorporated into a functional ycf1. The latter is possible, but given that ycf1 is a non-functional pseudogene in some other angiosperms, the likely explanation is that it is so here. Have the authors confirmed that ycf1 is an open reading frame in these plastomes? For that matter, ndhF is known to be a pseudogene in some organisms, too, but I haven’t heard of that being the case in any Lamiaceae, including Holmskioldia, for which an ndhF sequence was determined many years ago.

Minor notes:

P. 14, line 307. Why is this dinucleotide repeat labeled AT/AT, instead of AT/TA? All other repeats are labeled with their complement following the “/”. I think this should follow that convention and be AT/TA here and in any tables or figures.

P. 14, line 297; p. 15, line 331. On page 14 the total number of SSRs is put at 590; on page 15, it is 489.

P 14, lines 318-321. Many species names misspelled here.

Fig. 6. Why does sequence identity drop to zero at the 94k mark in the IRa of S. kingii and also at the 134k mark in the IRb of S. quadrilobata? In both places these are not matched in the corresponding other Inverted Repeat. Since they are in the IR, it doesn’t make sense that these are deletions.

P. 15, line 356. What does “conserved” mean if the actual boundary varies among species?

P. 15, line 360. “was detected in the IRa/SSC border region.” This is too vague. It is not just in the “border region,” but in the IRa.

Figure 5. The IR regions are depicted by a line above the rest of the figure. Since the boundary of the IRs vary among taxa, it should be noted in the caption which taxon these regions are specific to.

P. 22, Line 548. I encourage including the citation for Wagstaff et al., 1998 here, too, since they were first to show Tinnea as sister to Scutellaria (with Holmskioldia sister to them).

Signed: Richard Olmstead

7. PLOS authors have the option to publish the peer review history of their article (what does this mean?). If published, this will include your full peer review and any attached files.

Reviewer #2: Yes: Richard Olmstead

---

## [Author Response · Author response to Decision Letter 1]

12 Mar 2020

Major points:

1) Thanks to the authors for their reply to my query about the difference between forward, reverse, reverse complement, and palindromic repeats. It helped me to see the answer to a different concern I had raised regarding three of the SSRs that are not presented as complements in their paper. These are: AATC/ATTG, AAATC/ATTTG, and AAATAG/ATTTCT. I think these are labeled wrongly, with the mistake possibly being an improper computer output that was not caught by the authors, although I may still be mistaken in my understanding. Let’s work an example.

If AATC/ATTG represents a reverse complement repeat, the reverse complement of AATC would be GATT. In the genome the repeat would look like this: GATTGATTGATTGATTGATTGATTGATT etc.I think an error worked its way into the designation of this SSR by shifting the reverse complement repeat by one nt from GATT to ATTG. Functionally, of course, there is no difference. I think the proper way to present this in text and figures is as AATC/GATT, instead of AATC/ATTG, which does not make sense to me.

A similar adjustment will realign the other two SSRs as well:

AAATC/ATTTG should be AAATC/GATTT (offset by one nt)

AAATAG/ATTTCT should be AAATAG/CTATTT (offset by two nt)

As suggested by the reviewer, we have realigned three types of SSRs in the context, tables, and figures.

2) Figure 3 is a creative way to depict the numbers of different SSRs and how they are distributed among the taxa in the study. However, I don’t think the authors understood my concern about the shared ancestry of SSRs between related taxa The new Fig. 3 shows how many copies of each identified SSR occur is each taxon’s plastome and these are summed across the circle in the individual SSR. However, this doesn’t indicate in any way whether an individual SSR is shared due to common ancestry or not. If the authors have mapped each SSR, they should be able to identify how many unique SSR loci there are. I suspect that there are some that might be shared by all and some that are unique to a single taxon. The number of unique loci will fall somewhere between 48, the maximum number found in one plastome, and 489, the total number of SSRs summed across all plastomes. For my money, this is the most interesting cross-taxon assessment of their data that they could do, but if it is a lot of work to do, I can understand their reluctance to include it.

 Corrected. We provided a new figure to show which SSRs are shared by different species and which SSRs are unique to a specific species.

3) I think there is still a problem with the depiction of the Inverted Repeat/Small Single Copy regions in Fig. 6. At the IRa/SSC boundary, in Holmskioldia and several spp. of Scutellaria, the end of the IRa that is closest to the SSC is shown having both a portio of ycf1 and ndhF. If the IR has migrated into the SSC in these taxa, then I think the ycf1 pseudogene should be offset from the boundary by at least the amount of ndhF that is now found in the Ira. They can’t both occupy the same space. I understand that these are not drawn to scale (e.g., in Holmskioldia there are 1077 nt of ycf1 and only 39 nt of ndhF), but the figure still depicts something that does not make sense. Similarly for those same taxa, the same amount of ndhF will be found in the IRb, but is not depicted there in fig. 6. If ycf1 crosses the IRb/SSC boundary, but the IR boundary has migrated into ndhF, so that a portion (39 bp and perhaps some flanking DNA) of ndhF is now in the IRb, then ycf1 must either be a pseudogene in that location, too, or the ndhF DNA is incorporated into a functional ycf1. The latter is possible, but given that ycf1 is a non-functional pseudogene in some other angiosperms, the likely explanation is that it is so here. Have the authors confirmed that ycf1 is an open reading frame in these plastomes? For that matter, ndhF is known to be a pseudogene in some organisms, too, but I haven’t heard of that being the case in any Lamiaceae, including Holmskioldia, for which an ndhF sequence was determined many years ago.

Corrected. We have modified the location of pseudogene ycf1 (ψycf1) and the ndhF gene at the region of IRa/SSC boundary. This modification can show that the ndhF gene extended fractionally into the IRa region in some species (i.e., Holmskioldia sanguinea, Scutellaria altaica, S. amoena var. amoena, S. baicalensis, S. calcarata, S. insignis, S. kingiana, S. lateriflora, S. mollifolia, S. orthocalyx, S. przewalskii, S. quadrilobulata)

Minor notes:

P. 14, line 307. Why is this dinucleotide repeat labeled AT/AT, instead of AT/TA? All other repeats are labeled with their complement following the “/”. I think this should follow that convention and be AT/TA here and in any tables or figures.

 Actually, the repeat sequence before and after the “/” show different directions. Sequences before the “/” show a forward reading direction, while sequences after the “/” show a reversed reading direction. So, the repeat “AT/AT” can be explained like this: the AT before the “/” was read in the forward direction, while the AT after the “/” was read in the opposite direction. Then the dinucleotide repeat labeled as “AT/AT”.

P. 14, line 297; p. 15, line 331. On page 14 the total number of SSRs is put at 590; on page 15, it is 489.

 Corrected. On the page 14, the number 590 were the total number of the simple sequence repeats (SSR). While on page 15, the number 489 were the total numbers of the long repeat. In the revised version we corrected it “In total, 489 long repeats including forward, reverse, and palindromic were detected in the 15 plastomes…”

P 14, lines 318-321. Many species names misspelled here.

Corrected. See in the revised.

Fig. 6. Why does sequence identity drop to zero at the 94k mark in the IRa of S. kingiana and also at the 134k mark in the IRb of S. quadrilobata? In both places these are not matched in the corresponding other Inverted Repeat. Since they are in the IR, it doesn’t make sense that these are deletions.

 Corrected. The sequence identity of the 94k mark in the S. kingiana and 134k of the S. quadrilobata were not zero, but means that percentage of sequence identity below 50% which were calculated by the program mVISTA under the Shuffle-LAGAN model using a glocal alignment strategy. This time, we used LAGAN alignment model to reanalyze, and result were presented in Figure 5.

P. 15, line 356. What does “conserved” mean if the actual boundary varies among species?

 Corrected. Here, “conserved” mean that IRa/SSC boundary within Holmskioldia and Scutellaria spp. shows a little variation and no structural variation were detected. Now, we corrected it as “……. a small fragment of the ndhF gene extended into the IRa region with (29 bp in H. sanguinea and 25–45 bp among species of Scutellaria).

P. 15, line 360. “was detected in the IRa/SSC border region.” This is too vague. It is not just in the “border region,” but in the IRa.

 Corrected. 

Figure 5. The IR regions are depicted by a line above the rest of the figure. Since the boundary of the IRs vary among taxa, it should be noted in the caption which taxon these regions are specific to.

Corrected. We tried to use a line to show the general location of IR region. We have deleted this line because the boundary of IR regions varied among different species. 

P. 22, Line 548. I encourage including the citation for Wagstaff et al., 1998 here, too, since they were first to show Tinnea as sister to Scutellaria (with Holmskioldia sister to them).

 Corrected.

---

## [Decision Letter · Decision Letter 2]

20 Apr 2020

Leveraging plastomes for comparative analysis and phylogenomic inference within Scutellarioideae (Lamiaceae)

PONE-D-19-30248R2

Dear Dr. Xiang,

We are pleased to inform you that your manuscript has been judged scientifically suitable for publication and will be formally accepted for publication once it complies with all outstanding technical requirements.

With kind regards,

Genlou Sun

Academic Editor

PLOS ONE

Additional Editor Comments (optional):

Reviewers' comments:

Reviewer's Responses to Questions

**Comments to the Author**

1. If the authors have adequately addressed your comments raised in a previous round of review and you feel that this manuscript is now acceptable for publication, you may indicate that here to bypass the “Comments to the Author” section, enter your conflict of interest statement in the “Confidential to Editor” section, and submit your "Accept" recommendation.

Reviewer #2: All comments have been addressed

2. Is the manuscript technically sound, and do the data support the conclusions?

Reviewer #2: (No Response)

3. Has the statistical analysis been performed appropriately and rigorously? 

Reviewer #2: (No Response)

4. Have the authors made all data underlying the findings in their manuscript fully available?

Reviewer #2: (No Response)

5. Is the manuscript presented in an intelligible fashion and written in standard English?

Reviewer #2: (No Response)

6. Review Comments to the Author

Reviewer #2: (No Response)

7. PLOS authors have the option to publish the peer review history of their article (what does this mean?). If published, this will include your full peer review and any attached files.

Reviewer #2: Yes: Richard Olmstead

---

## [Editor Report · Acceptance letter]

23 Apr 2020

PONE-D-19-30248R2 

Leveraging plastomes for comparative analysis and phylogenomic inference within Scutellarioideae (Lamiaceae) 

Dear Dr. Xiang:

I am pleased to inform you that your manuscript has been deemed suitable for publication in PLOS ONE. Congratulations! Your manuscript is now with our production department. 

With kind regards,

on behalf of

Dr. Genlou Sun 

Academic Editor

PLOS ONE